# Deep Learning for Medical Image-Based Cancer Diagnosis

**DOI:** 10.3390/cancers15143608

**Published:** 2023-07-13

**Authors:** Xiaoyan Jiang, Zuojin Hu, Shuihua Wang, Yudong Zhang

**Affiliations:** 1School of Mathematics and Information Science, Nanjing Normal University of Special Education, Nanjing 210038, China; jxy@njts.edu.cn (X.J.); hzj@njts.edu.cn (Z.H.); 2School of Computing and Mathematical Sciences, University of Leicester, Leicester LE1 7RH, UK; shuihuawang@ieee.org

**Keywords:** cancer diagnosis, artificial intelligence, medical image, deep neural network, deep learning, overfitting prevention technique

## Abstract

**Simple Summary:**

Deep learning has succeeded greatly in medical image-based cancer diagnosis. To help readers better understand the current research status and ideas, this article provides a detailed overview of the working mechanisms and use cases of commonly used radiological imaging and histopathology, the basic architecture of deep learning, classical pretrained models, common methods to overcome overfitting, and the application of deep learning in medical image-based cancer diagnosis. Finally, the data, label, model, and radiomics were discussed specifically and the current challenges and future research hotspots were discussed and analyzed.

**Abstract:**

(1) Background: The application of deep learning technology to realize cancer diagnosis based on medical images is one of the research hotspots in the field of artificial intelligence and computer vision. Due to the rapid development of deep learning methods, cancer diagnosis requires very high accuracy and timeliness as well as the inherent particularity and complexity of medical imaging. A comprehensive review of relevant studies is necessary to help readers better understand the current research status and ideas. (2) Methods: Five radiological images, including X-ray, ultrasound (US), computed tomography (CT), magnetic resonance imaging (MRI), positron emission computed tomography (PET), and histopathological images, are reviewed in this paper. The basic architecture of deep learning and classical pretrained models are comprehensively reviewed. In particular, advanced neural networks emerging in recent years, including transfer learning, ensemble learning (EL), graph neural network, and vision transformer (ViT), are introduced. Five overfitting prevention methods are summarized: batch normalization, dropout, weight initialization, and data augmentation. The application of deep learning technology in medical image-based cancer analysis is sorted out. (3) Results: Deep learning has achieved great success in medical image-based cancer diagnosis, showing good results in image classification, image reconstruction, image detection, image segmentation, image registration, and image synthesis. However, the lack of high-quality labeled datasets limits the role of deep learning and faces challenges in rare cancer diagnosis, multi-modal image fusion, model explainability, and generalization. (4) Conclusions: There is a need for more public standard databases for cancer. The pre-training model based on deep neural networks has the potential to be improved, and special attention should be paid to the research of multimodal data fusion and supervised paradigm. Technologies such as ViT, ensemble learning, and few-shot learning will bring surprises to cancer diagnosis based on medical images.

## 1. Introduction

In recent years, the global incidence of cancer has remained high. Tens of millions of people are newly diagnosed with various types of cancer every year. At the same time, millions to nearly tens of millions of people around the world are killed by various types of cancer [1]. According to the 2020 global cancer burden data released by the International Agency for Research on Cancer (IARC) of the World Health Organization, the latest incidence and mortality trends of 36 cancer types in 185 countries worldwide are still grim. Based on the latest incidence rates, the world’s top ten cancers are female breast, lung, skin, prostate, colon, stomach, liver, rectum, esophageal, and cervix uteri [2].

At present, the diagnosis of cancer mainly depends on imaging diagnosis and pathological diagnosis [3,4]. In this case, early detection is the key to improving the survival rate of cancer patients [5]; non-invasive and efficient early screening has become an essential research topic. Imaging techniques include B-ultrasound, X-ray, computed tomography (CT), magnetic resonance imaging (MRI), etc. [6]. Through these imaging techniques, some cancerous symptoms of the body can be seen. A shadow in the lungs can be detected by CT, which can determine whether it is a symptom of lung cancer [7,8]. MRI is not only used to assist in the diagnosis and differentiation of nasopharyngeal carcinoma but also can be used to evaluate the extent of the cancer lesion: whether it involves the surrounding soft tissue and bone and whether there is metastasis to nearby lymph nodes [9]. Nodules or masses of different sizes in the thyroid can be found through a B-ultrasound examination and can also directly observe the size, shape, location, and boundary of the tumor in the thyroid through B-ultrasound [10]. Faced with a large amount of complex medical imaging information and growing demand for medical imaging diagnosis, artificial imaging diagnosis has many shortcomings, such as a heavy workload, susceptibility to subjective cognition, low efficiency, and high misdiagnosis rate.

Deep learning (DL), a branch of machine learning, is an algorithm based on an artificial neural network to learn features from data [11]. Deep learning proposes a method that enables computers to learn pattern features automatically and integrates feature learning into the process of model building, thus reducing the incompleteness caused by artificial design features and realizing the development of end-to-end prediction models [12].

Algorithms based on deep learning have advantages over humans in processing large data and complex non-deterministic data as well as in-depth mining of potential information in data [13]. Using deep learning to interpret medical images can help doctors locate lesions, assist in diagnosis, reduce the burden on doctors, reduce medical misjudgments, and improve the accuracy and reliability of diagnosis and prediction results. Deep learning techniques were successfully applied in various fields through medical images and physiological signals. Deep models have demonstrated excellent performance in many fields, such as medical image classification, segmentation, lesion detection, and registration [14,15,16]. Various types of medical images, such as X-ray, MRI, CT, etc., were used to develop accurate and reliable DL models to help clinicians diagnose lung cancer, rectal cancer, pancreatic cancer, gastric cancer, prostate cancer, brain tumors, breast cancer, etc. [17,18,19].

Due to the benefits of deep learning in cancer diagnosis, a large number of researchers are attracted. As the technology continues to improve, there is still spacious room for deep learning to be used in medical image analysis. In view of this, we review and summarize the development of deep learning in medical image-based cancer diagnosis through surveys, aiming to provide a more comprehensive introduction for beginners and to provide the latest deep learning techniques for researchers in this field.

We conducted a systematic review according to the Preferred Reporting Items for Systematic Review and Meta-Analysis (PRISMA). Through Google Scholar, Elsevier, and Springer Link, we searched for papers on the application of deep learning in medical image-based cancer diagnosis with the keywords “deep learning”, “cancer”, and “medical image”. Based on the title and content, we eliminated irrelevant and duplicate papers and finally included 389 papers.

In summary, the novelty and contributions of this work are as follows:(i)The principle and application of radiological and histopathological images in cancer diagnosis are introduced in detail;(ii)This paper introduces 9 basic architectures of deep learning, 12 classical pretrained models, and 5 typical methods to overcome overfitting. In addition, advanced deep neural networks are introduced, such as vision transformers, transfer learning, ensemble learning, graph neural network, and explainable deep neural networks;(iii)The application of deep learning technology in medical imaging cancer diagnosis is deeply analyzed, including image classification, image reconstruction, image detection, image segmentation, image registration, and image fusion;(iv)The current challenges and future research hotspots are discussed and analyzed around data, labels, and models.

The rest of this paper is organized as follows: In Section 2, we will introduce the common imaging techniques. In Section 3, we will show the basic architecture of deep learning, classical pretrained models, advanced deep neural networks, and overfitting prevention techniques. In Section 4, we will present a comprehensive review of the application of deep learning in medical image-based cancer diagnosis. In Section 5, we will describe the experimental details and a discussion of the results. The conclusion of this research will be described in Section 6.

## 2. Common Imaging Techniques

Currently, the most widely used medical images include radiology, pathology, endoscopy, etc. Radiological imaging techniques, such as X-ray, CT, MRI, B-ultrasound, etc., play an important role in the diagnosis of deep tissue lesions in the human body. The common radiological imaging techniques in cancer diagnosis are shown in Table 1 [20,21,22]. Pathological images refer to images of disease tissues and cells obtained by microscopy and other imaging techniques, including cytological and histological images.

### 2.1. Computed Tomography

Computed tomography (CT) is a technology that obtains three-dimensional reconstruction images of the inside of the human body by rotating and irradiating the human body with X-rays. The X-ray beam is used to scan the layer of a certain thickness of the human body. After receiving the X-ray through the layer, the detector converts it into visible light, which is converted into an electrical signal by the photoelectric converter, and then converted into a digital signal by an analog-to-digital converter and input to the computer for processing, and finally, the image is formed [42].

The basic principle of CT is shown in Figure 1 and the basic concepts and features of CT are shown in Table 2. According to the rays used, it can be divided into: X-ray CT (X-CT) and gamma-ray CT (γ-CT) [43]; the term “CT” usually refers to X-ray CT [44].

The four areas of most interest in CT are CT colonoscopy (virtual colonoscopy), CT lung screening of current and former smokers, CT cardiac screening, and CT whole-body screening [45]. Lung cancer screening with low-dose computed tomography has been shown to reduce mortality by 20–43% [46]. Tian et al. [47] developed a deep learning model to predict high PD-L1 expression in non-small cell lung cancer using computed tomography (CT) images and to infer the clinical outcome of immunotherapy. Body composition on preoperative chest computed tomography is an independent predictor of hospital length of stay (LOS) and postoperative complications after lobectomy for lung cancer [48]. Measuring body composition using diagnostic computed tomography (CT) has emerged as a method for assessing sarcopenia (low muscle mass) in cancer patients [49].

### 2.2. Magnetic Resonance Imaging

Magnetic resonance imaging (MRI) is the interaction of spinning nuclei with electromagnetic waves in a magnetic field to map the spatial position and related properties of specific nuclei or protons in an object [50]. The MRI system consists of four main parts: the main magnet, gradient system, radio frequency system, and computer system. The general imaging process is shown in Figure 2 and the basic concepts and features of MRI are shown in Table 3.

MRI has obvious advantages in diagnosing liver cancer and indeed the performance of liver cancer magnetic resonance is relatively specific. Liver cancer cases have typical imaging performance characteristics on magnetic resonance [51]. Magnetic resonance has a variety of sequences, each of which has a certain value in the diagnosis of liver cancer. Very small liver cancer lesions can be detected through magnetic resonance [52,53]. In addition, MRI has a high value in the diagnosis of other abdominal cancer, such as pancreatic cancer and kidney cell carcinoma. MRI has the potential as a biomarker for the diagnosis, treatment, and prognosis of renal cell carcinoma (RCC) [54,55].

MRI has been shown to be equal to or superior to other imaging modalities in the diagnosis of pancreatic cancer and its versatility can accurately delineate the pancreatic parenchyma as well as ductal structures, such as enhanced MRI, which can help distinguish pancreatic cancer from focal pancreas inflammation [56,57]. MRI has its unique superiority in examining blood vessels, which can be examined in different parts according to different coils. MRI is also the most commonly used examination for the diagnosis of brain tumors [58], providing detailed information on brain tumor anatomy, cellularity, and vascular supply [59]. MRI, especially dynamic contrast-enhanced MRI, is currently considered the most sensitive method for detecting breast cancer without ionizing radiation and an effective screening alternative for high-risk populations [60].

MRI has a high diagnostic value for hypertrophy tumors [61] and lung cancer staging [62,63]. MRI is significantly better than CT in the diagnosis of the staging of bladder cancer, prostate cancer, and cervical cancer [64,65]. Wu et al. [66] developed a deep-learning model to diagnose cervical cancer lymph node metastasis using magnetic resonance imaging before surgery. Multiparametric magnetic resonance imaging (mpMRI) can accurately perform local staging, predict tumor aggressiveness, and monitor response to therapy for bladder cancer (BCa) patients [67]. Contrast-enhanced MR and diffusion-weighted MR (DW-MRI) can distinguish non-muscle-invasive bladder cancer from muscle tumors with 80% accuracy [68]. mpMRI is a routine imaging method for the diagnosis of prostate cancer but 10–20% of prostate tumors are missed [69]. MRI in combination with prostate-specific membrane antigen positron emission tomography can reduce false negatives in prostate cancer (csPCa) [70].

The good tissue resolution of MRI can better demonstrate the invasive changes in the rectal wall and surrounding infiltration, especially the high-resolution magnetic resonance imaging (HRMRI) [71,72,73], intravoxel incoherent motion diffusion-weighted imaging (IVIMDWI) [74,75], dynamic contrast-enhanced magnetic resonance imaging (DCE-MRI) [76,77], diffusional kurtosis imaging (DKI) [78,79], and magnetic resonance spectroscopy (MRS) [80]; other imaging techniques have made MRI a comprehensive assessment of effective imaging methods of rectal cancer clinical diagnosis and treatment of rectal cancer [81].

**Table 3 cancers-15-03608-t003:** The basic concepts and features of MRI.

MRI	Description
Conception	◆ Contrast agents can be injected intravenously into patients before or during MRI to increase the velocity at which protons coincide with the magnetic field [82,83]; ◆ The brain, spinal cord, nerves, muscles, ligaments, and tendons can be seen more clearly with MRI than with conventional X-rays and CT [84].
Feature	◆ It can directly make tomographic images of cross-section, sagittal plane, coronal plane, and various oblique planes without producing artifacts in CT detection and has imaging parameters several times more than CT and high soft tissue resolution [85] in bones [86,87,88], joints (such as rheumatoid arthritis, tenosynovitis, synovitis) [89,90], and soft tissue lesions [91]. That is to say, subtle results that other imaging techniques cannot distinguish are displayed; ◆ Magnetic resonance imaging has no ionizing radiation [92]. Although it is recognized as a relatively safe imaging method, the human body must be placed in a huge static magnetic field during imaging. There are also potential security risks [93]; ◆ The imaging speed of MRI is slow, the examination time is long, and the patient’s voluntary or involuntary activities can cause motion artifacts and affect the diagnosis [94]; ◆ Its spatial resolution is not as good as that of CT, MRI cannot examine patients with cardiac pacemakers or parts with certain metal foreign bodies, and the price is relatively high.

### 2.3. Ultrasound

The sound waves with more than 200,000 vibrations per second are not felt by ears and are called ultrasonic. Ultrasound (US) can spread in a certain direction and penetrate objects. If an obstacle is encountered, echo will occur. Taking real-time two-dimensional color flow ultrasounds as an example, the imaging system block diagram is shown in Figure 3 [95] and the basic concepts and features of ultrasound are shown in Table 4.

Through B-ultrasound, a variety of clear sectional graphs of internal organs of the human body can be obtained. Most tumors can also be detected, such as breast cancer, thyroid cancer, liver cancer, pancreatic cancer, bladder cancer, kidney cancer, ovarian cancer, etc. [96,97,98]. Still, B-ultrasound cannot detect some cancers, such as bone tumors, gastrointestinal tumors, and lung tumors. As a result of the fact that the lungs and gastrointestinal tract contain gas, ultrasonic waves cannot penetrate the lungs and gastrointestinal tract due to reflection against the gas during the examination, so the detection of lung cancer and gastrointestinal tumors is limited [99]. Bone tumors cannot be penetrated by ultrasound when passing through bone, so it also greatly influences the examination of bone tumors [99].

Safe and effective delivery of anticancer drugs to lesions is one of the grand challenges in cancer therapy. An ultrasound-triggered drug delivery system (UTDDS) was developed into a new type of efficient and non-invasive drug delivery technology [97,100], such as using ultrasound to break through the blood–brain barrier to deliver drugs to brain tumors [101]. It can position ultrasound radiation in the tumor area and promote the regular and quantitative release of the drug through ultrasound stimulation to achieve a high concentration of local chemotherapy drugs and improve the anti-cancer effect. The thermal effect produced by high-intensity ultrasound is helpful for cancer treatment [102,103].

**Table 4 cancers-15-03608-t004:** The basic concepts and features of ultrasound.

Ultrasound	Description
Conception	◆ Different obstacles will produce different echoes. This echo is collected by instruments and displayed on a screen to understand the object’s internal structure; ◆ Ultrasound is used to diagnose and treat human diseases [104]. Many types of ultrasonic diagnostic instruments are used in clinical medicine, such as A-mode, B-mode, M-mode, and Doppler ultrasound-mode [105].
Feature	◆ B-mode ultrasonography is the most widely and convenient clinical application, without radiation, non-invasive, and suitable for most people.

### 2.4. X-ray

X-ray is a kind of electromagnetic wave with a very short wavelength. The wavelength is about 10 nm to ~0.1 nm. A typical X-ray imaging process is shown in Figure 4 and the basic concepts and features of X-ray are shown in Table 5.

X-ray is mainly used to detect bone lesions. Generally, X-ray images of bone in cancer areas differ from those of surrounding healthy bone and muscle areas, providing a low-cost screening tool for diagnosing and visualizing bone cancer [106]. X-rays are also useful for detecting soft tissue lesions. For example, chest X-rays (CXR) diagnose lung diseases such as pneumonia, lung cancer, or emphysema [107]. Abdominal X-rays detect intestinal infarcts, free gas, and free fluid. The sensitivity of chest X-rays for symptomatic lung cancer is only 77% to 80% [108,109]. Chest X-rays with artificial intelligence can capture more lung cancers [110,111].

**Table 5 cancers-15-03608-t005:** The basic concepts and features of X-ray.

X-ray	Description
Conception	◆ It has a strong penetrating ability to clearly show the internal structure of compressed objects after scanning and is used for non-destructive security checks and medical imaging [112]; ◆ The application of X-rays in medical diagnosis is mainly based on the penetration, differential absorption, photosensitivity, and fluorescence of X-rays [113].
Feature	◆ The human body is absorbed to different degrees by X-ray: the amount of X-ray absorbed by the bone is more than that absorbed by the muscle; the amount of X-ray after passing through the human body is different, which carries the information of the density distribution of each part of the human body; and the intensity of fluorescence or photosensitivity caused on the fluorescent screen or photographic film is greatly different. Thus, shadows of different densities will be displayed on the fluorescent screen or on the photographic film (after developing and fixing).

### 2.5. Positron Emission Tomography

Positron emission tomography (PET) imaging utilizes the β+ decay of radioactive isotopes, allowing noninvasive imaging [114], and is well suited for monitoring early cellular/molecular events during disease processes, as well as during drug or radiation therapy [115]. The process of PET imaging mainly includes detection data generation, data acquisition and storage, data processing and reorganization, image reconstruction, and image display and processing, as shown in Figure 5. The basic concepts and features of PET are shown in Table 6.

Comprehensive PET–CT can improve the accuracy of the staging diagnosis of non-small cell lung cancer (NSCLC) [116]. Despite the risk of false-positive bone lesions, PET–CT outperforms all other imaging modalities in detecting primary distant metastases from high-risk prostate cancer [117] and its role in lung cancer treatment has expanded exponentially [118]. MRI in combination with prostate-specific membrane antigen positron emission tomography can reduce false negatives for clinically significant prostate cancer (csPCa) compared with MRI, potentially reducing the number of prostate biopsies required to diagnose csPCa [70].

**Table 6 cancers-15-03608-t006:** The basic concepts and features of PET.

PET	Description
Conception	◆ PET is particularly valuable in diagnosing and treating three major diseases: tumors, coronary heart disease, and brain diseases; ◆ Most malignant tumors have high glucose metabolism; 2-[18F]fluoro-2-deoxy-D-glucose (FDG), as a compound similar to glucose in structure, will accumulate in malignant tumor cells after intravenous injection. Therefore, PET can distinguish malignant tumors from benign tumors and normal tissues and recurrent tumors from surrounding necrosis and scar tissue [115]; ◆ Currently, it is commonly used in the examination of lung cancer, breast cancer, colorectal cancer, ovarian cancer, lymphoma, melanoma, etc. [119,120]. PET can be used to judge whether the malignant tumor has metastasized and the location of the metastases, which plays an important role in guiding the staging of tumor diagnosis, whether surgery is required, and the scope of surgical resection [120].
Feature	◆ PET imaging only shows the location and concentration of radioactive isotopes but cannot image the structure of human tissue and PET does not provide precise information on the exact location of focal abnormalities [116,121]. Therefore, PET is usually used together with CT or MRI to achieve complementary advantages and improve the accuracy of disease diagnosis.

### 2.6. Histopathology

Histopathology refers to observing the morphological changes of histopathology cells under a microscope after biopsy or when surgical specimens are made into tissue slides to diagnose disease [122]. The generation of histopathological images mainly includes histopathological image acquisition, tissue slide, chemical staining, and histopathological image analysis [123]. The basic concepts and features of histopathological images are shown in Table 7.

## 3. Deep Learning

### 3.1. Basic Model

The concept of deep learning originates from the research of artificial neural networks [127]. A multi-layer perceptron with multiple hidden layers is a deep learning structure [128]. The commonly used models for deep learning mainly include convolutional neural network (CNN), deep belief network (DBN), deep autoencoder (DAE), restricted Boltzmann machine (RBM), etc. The basic deep learning architectures are shown in Table 8.

#### 3.1.1. Convolutional Neural Network

Using a convolutional neural network (CNN) [129] is the most prominent technique of deep learning in the field of image processing. CNN is mostly used for medical image processing tasks among the different neural network architectures [138,139]. It is a kind of feed-forward neural network with convolution calculation and a deep structure. It is specially designed to process neural networks with similar grid structure data, such as time series and image data. CNNs can be trained using the backpropagation algorithm. The main structure of a convolutional neural network includes the input layer, convolutional layer, pooling layer, fully connected layer, and output layer [140], as shown in Figure 6.

The purpose of the convolution operation is to extract different input features [141]. Some convolutional layers may only extract low-level features such as edges, lines, and corners. More layers of networks can iteratively extract more complex features from low-level features. The convolutional layer plays the role of feature extraction but it does not reduce the number of features of the picture.

The final fully connected layer still faces many parameters, so the pooling layer is required to reduce the number of features. The pooling layer mainly performs subsampling processing on the feature map learned by the convolutional layer, mainly consisting of two types: max pooling and average pooling [142]. The former takes the maximum value within the window as the output, while the latter takes the mean of all the values within the window as the output. The pooling layer reduces the input dimension of subsequent network layers, reduces the model’s size, improves the calculation speed, and improves the feature map’s robustness to prevent overfitting. Finally, the fully connected layer is used to realize recognition and judgment.

CNN combines local perception, weight sharing, and down sampling to fully use the locality and other characteristics of the data [143], optimize the network structure, and ensure a certain degree of invariance in displacement and deformation.

#### 3.1.2. Fully Convolutional Network

A fully convolutional network (FCN) is a framework for image semantic segmentation proposed by Shelhamer et al. [130] in 2015. It replaces the fully-connected layer behind the traditional CNN with a convolutional layer. It is a fully-convolutional network without a fully-connected layer which realizes pixel-level classification of images, thereby solving the semantic-level image segmentation problem.

The FCN network structure is mainly divided into two parts: the full convolution part and the deconvolution part [144]. The full convolution part entails some classic CNN networks (such as VGG, ResNet, etc.) used to extract features. Deconvolution is introduced into the up sampling process to up sample the feature map of the last convolutional layer to restore it to the same size as the input image so that a prediction can be generated for each pixel while retaining the spatial information in the original input image [145]. Finally, pixel classification is performed on the up sampled feature map and the output is a picture that has been labeled. Therefore, the input of FCN can be a color image of any size.

FCN uses the corresponding relationship between the output result and the input image, directly gives the classification of the corresponding region of the input image and cancels the sliding window selection candidate box in the traditional target detection. FCN itself still has many limitations. For example, it does not consider global information, cannot solve the problem of instance segmentation, is not sensitive to the details in the image, and the speed is far from real-time [146,147].

#### 3.1.3. Autoencoder

The autoencoder (AE) concept was first proposed by Rumelhart et al. [131]. Using an autoencoder entails an unsupervised data compression and data feature expression method. It is based on the backpropagation algorithm and optimization methods (such as the gradient descent method), using the input data as supervision to guide the neural network to try to learn a mapping relationship to obtain a reconstructed output of the original input.

The autoencoder can be seen as a three-layer network (input layer, hidden layer, and output layer), including an encoder and a decoder. The encoder maps the input data of the high-dimensional space to the encoding of the low-dimensional space to achieve data compression and the decoder decompresses the input data to achieve recurrence [148]. The decoder is usually eliminated and the encoder model is retained after completing the training phase to extract the input data features [149]. The structure of the autoencoder is shown in Figure 7.

In the encoding stage, the input x is encoded by the encoding function, as shown in Equation (1), and the encoding h is obtained to realize data compression.
(1)$$h=f\left(x\right)=\delta_{f}\left(Wx+a\right),$$
where $\delta_{f}$ is the activation function of the encoder, which is usually taken as sigmoid function, as shown in Equation (2).
(2)$$\delta_{f}\left(z\right)=\frac{1}{1+e^{-z}},$$
where W and a represent the weight matrix and bias vector of the encoder (i.e., between the input layer and the hidden layer), respectively.

In the decoding stage, the decoding function is used to map the code h to the original high-dimensional space to achieve the reproduction of the input, as shown in Equation (3):(3)$$y=g\left(h\right)=\delta_{g}\left(W^{T}x+b\right),$$
where $W^{T}$ and b are the weight matrix between the hidden layer and the output layer (for simplicity, the weight matrix here is taken as the transpose of W), $\delta_{g}$ is the activation function of the decoder, usually taken as the sigmoid function or the identity function, as shown in Equations (4) and (5), respectively.
(4)$$\delta_{g}\left(z\right)=\frac{1}{1+e^{-z}}$$
(5)$$\delta_{g}\left(z\right)=z$$

Single-layer autoencoders are not enough to obtain a good data representation due to their simple shallow structural features [150]. We can use deeper neural networks to better capture the semantic information of the data and improve its representation ability. A stacked autoencoder (SAE) generally adopts layer-wise unsupervised pre-training to learn network parameters. The essence of SAE is to increase the number of hidden layers on the basis of AE to obtain a better feature extraction capability. Due to the hierarchical structure, one of the most important features of SAE is that highly nonlinear and complex patterns can be learned or discovered [150].

When the sample has a sparse representation, the performance of the classification task will be improved [127]. The sparse autoencoder is obtained by adding some sparsity constraints on the basis of the traditional autoencoder. This sparsity is for the hidden layer neurons of the autoencoder. By suppressing most of the output of the hidden layer neurons, the network can achieve a sparse effect so that few useful feature items can be obtained. A neuron is considered active when the output of the neuron is close to 1 and it is considered inhibitory when the output is close to 0. The neuron is limited by our desire to be inactive most of the time [151].

In addition to properties such as the minimum reconstruction error or sparsity, effective data representation may require other properties, such as robustness to partial destruction of data. A denoising autoencoder (DAE) is an autoencoder that accepts damaged data as an input and trains them to predict the original undamaged data as an input [152]. The core idea of DAE is to increase the robustness of data encoding by introducing noise and improving the model’s generalization ability.

There are many other variants of autoencoders, such as a variational autoencoder (VAE) [153], contraction autoencoder (CAE) [154], etc. The common variants of autoencoder are shown in Table 9. Yousuff et al. [149] proposed a mixed-dimensional reduction method based on a deep automatic encoder for a machine-learning model of a melanoma cancer diagnosis. In this method, the Neighborhood Component Analysis (NCA) technique is applied to transform the 2090-dimensional spectral features into a 2090-dimensional vector and the deep autoencoder model is used to process non-linearity in the data and generate potential space. Xu et al. [155] applied a stacking sparse autoencoder (SSAE) for nuclei detection on high-resolution histopathological images of breast cancer. Stacked sparse autoencoder models can capture high-level feature representations of pixel intensities. A single-layer sparse autoencoder was successfully used to extract advanced features in the diagnosis of clinically significant prostate cancer (PCa) using multi-parameter magnetic resonance imaging (mpMRI) biomarkers in highly unbalanced datasets [156]. 

For cancer survival prediction, Huang et al. [157] used a typical autoencoder structure to reconstruct the original input, adopted sparse coding technology to optimize the network structure, and minimized the number of network weights to identify information features accordingly so that the optimal features were selected and the generalization ability was enhanced. Munir et al. [158] designed a sparse deep convolutional autoencoder model to assist breast cancer diagnoses. The input image in this model can be the low-rank, multi-channel, and approximate thermal base.

**Table 9 cancers-15-03608-t009:** Comparison of commonly used variant autoencoders.

Author	Method	Year	Description	Feature
Bengio et al. [159]	SAE ^1^	2007	Use layer-wise training to learn network parameters.	The pre-trained network fits the structure of the training data to a certain extent, which makes the initial value of the entire network in a suitable state, which is convenient for the supervised stage to speed up the iterative convergence.
Vincent et al. [160]	DAE ^2^	2008	Add random noise perturbation to the input data.	Representation reconstructs high-level information from chaotic information, allowing high learning capacity while preventing learning a useless identity function in the encoder and decoder, improving algorithm robustness, and obtaining a more efficient representation of the input.
Vincent et al. [152]	SDAE ^3^	2010	Multiple DAEs are stacked together to form a deep architecture. The input is corroded (noised) only during training; once the training is complete, there is no need to corrode.	It has strong feature extraction ability and good robustness. It is just a feature extractor and does not have a classification function.
Ng [151]	Sparse autoencoder	2011	A regular term controlling sparsity is added to the original loss function.	Features can be automatically learned from unlabeled data and better feature descriptions can be given than the original data.
Rifai et al. [154]	CAE ^4^	2011	The autoencoder object function is constrained by the encoder’s Jacobian matrix norm so that the encoder can learn abstract features with anti-jamming.	It mainly mines the inherent characteristics of the training samples, which entails using the gradient information of the samples themselves.
Masci et al. [161]	Convolutional autoencoder	2011	Utilizes the unsupervised learning method of the traditional autoencoder, combining the convolution and pooling operations of the convolutional neural network.	Through the convolution operation, the convolutional autoencoder can well preserve the spatial information of the two-dimensional signal.
Kingma et al. [153]	VAE ^5^	2013	Addresses the problem of non-regularized latent spaces in autoencoders and provides generative capabilities for the entire space.	It is probabilistic and the output is contingent; new instances that look like input data can be generated.
Srivastava et al. [162]	Dropout autoencoder	2014	Reduce the expressive power of the network and prevent overfitting by randomly disconnecting the network.	The degree of overfitting can be reduced and the training time is long.
Srivastava et al. [163]	LAE ^6^	2015	Compressive representations of sequence data can be learned.	Representation helps improve classification accuracy, especially when there are few training examples.
Makhzani et al. [164]	AAE ^7^	2015	An additional discriminator network is used to determine whether hidden variables of dimensionality reduction are sampled from prior distributions.	Minimize the reconstruction error of traditional autoencoders; match the aggregated posterior distribution of the latent variables of the autoencoder with an arbitrary prior distribution.
Xu et al. [155]	SSAE ^8^	2015	Advanced feature representations of pixel intensity can be captured in an unsupervised manner.	Only advanced features are learned from pixel intensity to identify the distinguishing features of the kernel; efficient coding can be achieved.
Higgins et al. [165]	beta-VAE	2017	beta-VAE is a generalization of VAE that only changes the ratio between reconstruction loss and divergence loss. The scalar β denotes the influence factor of the divergence loss.	The potential channel capacity and independence constraints can be balanced with the reconstruction accuracy. Training is stable, makes few assumptions about the data, and relies on tuning a single hyperparameter.
Zhao et al. [166]	info-VAE	2017	The ELBO objective is modified to address issues where variational autoencoders cannot perform amortized inference or learn meaningful latent features.	Significantly improves the quality of variational posteriors and allows the efficient use of latent features.
Van Den Oord et al. [167]	vq-VAE ^9^	2017	Combining VAEs with vector quantization for discrete latent representations.	Encoder networks output discrete rather than continuous codes; priors are learned rather than static.
Dupont [168]	Joint-VAE	2018	Augment the continuous latent distribution of a variational autoencoder using a relaxed discrete distribution and control the amount of information encoded in each latent unit.	Stable training and large sample diversity, modeling complex continuous and discrete generative factors.
Kim et al. [169]	factorVAE	2018	The algorithm motivates the distribution of the representation so that it becomes factorized and independent in the whole dimension.	It outperforms β-VAE in disentanglement and reconstruction.

^1^ SAE: Stacked autoencoder; ^2^ DAE: Denoising autoencoder; ^3^ SDAE: Stacked denoising autoencoder; ^4^ CAE: Contractive autoencoder; ^5^ VAE: Variational autoencoder; ^6^ LAE: LSTM autoencoder; ^7^ AAE: Adversarial autoencoder; ^8^ SSAE: Stacked sparse autoencoder; ^9^ vq-VAE: Vector quantized-variational autoencoder.

#### 3.1.4. Deep Convolutional Extreme Learning Machine

The extreme learning machine (ELM) is a fast learning algorithm proposed by Huang et al. [170] which can recognize images with a single hidden layer neural network. During the training, there is no need to adjust or update parameters, only the hidden layer nodes are adjusted to find the optimal solution [171]. Compared with traditional classification methods such as CNN and SVM [172,173], ELM has a faster learning speed and stronger generalization performance. However, feature learning using ELM methods may not be effective for some image classification applications due to their shallow architecture.

Taking advantage of the corresponding network, the feature extraction performance of the convolutional neural network and the fast training of the extreme learning machine are combined in a deep convolutional extreme learning machine (DC-ELM) [132]. As shown in Figure 8, the structure of DC-ELM consists of an input layer, an output layer, and multiple hidden layers. The hidden layers are alternately arranged as convolutional and pooling layers which can effectively extract high-level features from the input image. The convolutional layer is composed of several feature maps. The same feature map shares the same input weight but the input weight of different feature maps differs. The pooling layer uses the square root pooling layer to enable the network to have frequency selective and translation invariance. It has the same number and size of feature maps as the previous convolution layer. The last layer is fully connected with the output layer and adopts the random pooling strategy to reduce the size of its feature graph and save significant computing resources.

#### 3.1.5. Recurrent Neural Network

A recurrent neural network (RNN) is a class of recursive neural networks that takes sequence data as the input, makes recursion in a sequence’s progression direction, and links cyclic units in a chain. RNN is suitable for dealing with timing-related issues such as video, voice, and text. In the field of medical images, an RNN is used to assist in cancer classification and assessment [174,175,176,177], prediction [178,179], and clinical data modeling [180]. In order to achieve better accuracy, RNNs are often combined with other networks (such as convolutional cyclic neural network [177,181], convolutional grid neural network [182], hybrid method of the recurrent neural network and graph neural network (RGNN) [183], and wavelet recurrent neural network [184]), or the internal structure of RNNs can be improved (such as bidirectional recurrent neural network [178,185], long short-term memory network (LSTM) [186], and fuzzy recurrent neural network (FR–Net) [187]).

In RNNs, the current output of a sequence is also related to the previous output. The specific manifestation is that the network will remember the previous information and apply it to calculate the current output. That is, the nodes between the hidden layers are connected and the input of the hidden layer includes the output of the input layer and the output of the previously hidden layer. In theory, RNNs can process sequence data of any length. However, it is often assumed that the current state is only related to the previous states to reduce complexity in practice. The model structure is shown in Figure 9.

x, s, and y are all vectors representing the values of the input layer, hidden layer, and output layer, respectively. O is the weight matrix from the input layer to the hidden layer and P is the weight matrix from the hidden layer to the output layer. The weight matrix Q is the last value of the hidden layer as the input weight of this time.

After the network receives the input $x_{t}$ at time t, the value of the hidden layer is $s_{t}$ and the output value is $y_{t}$. The value of $s_{t}$ depends not only on $x_{t}$, but also on $x_{t-1}$. We can use the following formula to express the calculation method of the cyclic neural network:(6)$$y_{t}=g\left(P\cdot s_{t}\right)$$
(7)$$s_{t}=f\left(O\cdot x_{t}+Q\cdot s_{t-1}\right)$$

It should be noted that in the same hiding layer, O, P, and Q at different times are all equal, which is the parameter sharing of RNN. These parameters are updated when backpropagation is performed.

The parameter learning of the cyclic neural network can be learned through the backpropagation algorithm over time. That is, the error is passed forward step by step in the reverse order of time. Unfortunately, when the input sequence is relatively long, the gradient explosion or gradient disappearance will occur, which is also called the long-term dependence problem. Gating mechanisms are introduced to improve recurrent neural networks to solve this problem.

#### 3.1.6. Long Short-Term Memory

Long and short-term memory (LSTM) is a kind of RNN proposed by Hochreiter et al. [134] in 1997, which is suitable for processing and predicting important events with relatively long intervals and delays in time series. It is used to solve the problem of explosion or disappearance that usually occurs due to the long-term dependence of RNN [188]. Gers et al. [189] added “Forget Gate” to LSTM in 2000, allowing the network to reset its state in order to strengthen the proper memory reset function. Forgetting may occur rhythmically or in an input-dependent manner. An ordinary LSTM cell consists of a cell, an input gate, an output gate, and a forget gate.

The ability of the LSTM to retain long-term information lies in the design of the “gate” structure. In LSTM, the first stage is the forget gate which determines what information needs to be forgotten from the cell state. The next stage is the input gate, which determines what new information can be stored in the cell state. And the last stage is the output gate, which determines what values to output.

Medical image data acquisition time points will appear uneven and irregular. In order to model both regular and irregular longitudinal samples, LSTM networks are widely used in serial observation modeling in fields such as medical imaging [190]. Researchers have shown that LSTM is superior to other models in processing irregular medical data [191,192]. Koo et al. [193] developed an online decision support system based on the LSTM model for survival prediction of prostate cancer, providing individualized survival outcomes based on initial treatment plans. Time-series tumor marker (TM) data were used to further improve the predictive performance of LSTM models, even with widely varying intervals between tests, so that occult tumors could be detected earlier [191].

#### 3.1.7. Generative Adversarial Network

A generative adversarial network (GAN) was systematically proposed by Goodfellow et al. [135]. The GAN network structure contains two models: one is the generator and the other is the discriminator. During the training process, both the real data extracted from the dataset and the fake data that the generator keeps creating throughout the training process are included [194]. The generator is used to capture the distribution of the entire data. The generator fits and approximates the real data distribution; the discriminator estimates whether a sample comes from real data or is generated. The GAN model structure diagram is shown in Figure 10.

We input random noise n into the generator G to generate samples $x_{F}=G\left(n\right)$, the function G is just a function represented by a neural network, which converts random and unstructured n vectors into structured data with the aim of being statistically indistinct from the training data.

The data satisfying the real distribution is recorded as $x_{R}$. $x_{F}$ and $x_{R}$ are sent to the discriminator D at the same time for training. The discriminator is trained in much the same way as any other binary classifier, except that the pseudo-classes data come from a constantly changing distribution as the generator learns rather than from a fixed distribution. The generator and the discriminator form a mutual confrontation relationship. During repeated game training, the recognition ability of the network becomes stronger and stronger. Finally, the network reaches the optimal state when the discriminator cannot distinguish whether the generated result is true (discrimination probability is 0.5).

In the application of medical images to assist in cancer diagnosis, GAN can be used to generate medical images that are identical to real images to solve the problem of insufficient training data [195].

#### 3.1.8. Deep Belief Network

The deep belief network (DBN) was proposed by Geoffrey Hinton in 2006 [136]. It is an effective method to solve the problems of slow learning speed and the overfitting phenomenon of deep neural networks [196]. It is a probabilistic generation model that gives the entire network a better initial weight through layer-by-layer training to reach the optimal solution after fine-tuning [136,197].

The restricted Boltzmann machine (RBM) plays the most important role in layer-by-layer training, which is a two-layer neural network. The first layer is a visible layer representing data used to input training data. The second layer is a hidden layer of hidden units representing features that capture high-order correlations in the data, used as feature detectors [198]. Adjacent layers of RBM are connected. Each neuron in the visible layer is connected to all neurons in the hidden layer but there is no connection between neurons in the same layer and the output state of all neurons is only two kinds [199].

DBN is a neural network composed of multi-layer RBM. It can be regarded as either a generation model or a discriminant model. Its training process is to pre-train to obtain weights by using an unsupervised greedy layer-by-layer method. The classic DBN network structure is a deep neural network composed of several layers of RBM and a layer of BP, as shown in Figure 11 [196].

DBN is mainly divided into two steps in the process of training the model [196].

Step 1 pre-training: train each layer of the RBM network separately and unsupervised to ensure that feature information is retained as much as possible when feature vectors are mapped to different feature spaces.

Step 2 fine-tuning: set the BP network in the last layer of DBN, receive the output feature vector of RBM as its input feature vector, and train the entity relationship classifier under supervision. And each layer of the RBM network can only ensure that the weight value is optimal for the feature vector mapping of this layer, not for the feature vector mapping of the entire DBN, so the backpropagation network also propagates error information from top to bottom to each layer of RBM, thus fine-tuning the entire DBN network.

The process of the RBM network training model can be regarded as the initialization of weight parameters of a deep BP network so that DBN overcomes the shortcomings of the BP network that are easy to fall into local optimum and long training times due to the random initialization of weight parameters [200]. The top layer of the above training model with supervised learning can be replaced with any classifier model according to the specific application field instead of the BP network. By training the weights between its neurons, the entire neural network can generate training data according to the maximum probability. We can not only use DBN to identify features and classify data but also use it to generate data to solve the problem of insufficient sample size [201].

The application of deep belief networks involves two main challenges: the method of fine-tuning the network weights and biases and the number of hidden layers and neurons [202]. Ronoud et al. [202] applied extreme learning machine (ELM) and backpropagation (BP) algorithms to DBN to optimize the fine-tuning method of network weights and used the genetic algorithm to optimize the DBN structure in the proposed model to correctly give the number of hidden layers in DBN and the number of neurons in each layer.

#### 3.1.9. Deep Boltzmann Machine

The deep Boltzmann machine (DBM) is a deep learning model based on the restricted Boltzmann machine, proposed by Salakhutdinov et al. [137]. It is essentially composed of stacked multi-layer restricted Boltzmann machines. Uniquely from the deep belief network, the middle layer of the DBM is bidirectional and connected to the adjacent layer. Compared with RBM and DBN, DBM has a powerful shape representation ability [203] which can capture more global and local constraints of object shape so as to construct a strong probabilistic object shape model that meets the requirements of realism and generalization.

Wu et al. [204] built a heart shape model by training a three-layer DBM to characterize local and global heart shape variations for better heart motion tracking. Syafiandini et al. [205] used the multimodal deep Boltzmann machine to learn important genes (biomarkers) on gene expression data of human colorectal cancer, including several patient phenotypes, such as the occurrence of lymph nodes and distant metastases. Hess et al. [206] used deep Boltzmann machines to model immune cell gene expression patterns in lung adenocarcinoma and partitioned DBMs to simulate the expression of immune cell-related genes in lung adenocarcinoma. Their approach solves the problem that DBMs are limited to the amount of individual training data far greater than the amount of feature data.

### 3.2. Classical Pretrained Model

This section will introduce typical deep learning architectures based on neural networks, the basics of which are shown in Table 10 [207].

#### 3.2.1. LeNet-5

The LeNet-5 was proposed by Lecun et al. [129] in 1998 to solve the visual task of handwritten character recognition. The convolutional network architecture is based on the following three ideas: local receptive field, shared weights, and sub-sampling in time or space, which is the latter pooling layer, to ensure a certain degree of invariance to shift, scale, and distortion.

LeNet-5 has a total of seven layers. In addition to the input layer, each layer contains trainable parameters. Each network layer generates multiple feature maps and each feature map can extract a type of feature of the input data through a convolution filter. The LeNet network includes the basic modules of deep learning: convolution, pooling, and fully connected layers, as shown in Figure 12 [129]. Feature extraction is performed through operations such as convolution and pooling, and finally, classification and recognition are realized using full connections.

The weight-sharing feature of the convolutional layer saves a considerable amount of calculation and memory space compared with the fully connected layer. In the convolution operation, neurons are locally connected in spatial dimension but fully connected in depth. For the two-dimensional image, the local pixel correlation is strong and the local connection feature ensures that the learned filter can respond strongly to the local input features.

#### 3.2.2. AlexNet

AlexNet [208] deepens the network structure on the basis of LeNet and learns richer and higher-dimensional image features. Compared to LeNet, AlexNet has a deeper network structure with five convolutional layers and three fully connected layers. The output of the last fully connected layer is fed to a 1000-way softmax which produces a distribution over the 1000 class labels.

In AlexNet, the rectified linear unit (ReLU) is used as the activation function of CNN which successfully solves the problem of gradient dispersion of the sigmoid function when the network is deep [223]. In addition, the ReLU function is simpler than the sigmoid function and requires less computation, which speeds up the training speed. During the training process of the neural network, the powerful parallel computing capability of the GPU is used to handle a large number of matrix operations to speed up the training. AlexNet distributes parallel training on two GPUs, stores half of the neuron parameters in the video memory of each GPU, and the GPU only communicates in specific layers [224].

The local response normalization (LRN) layer was proposed to create a competition mechanism for the activity of local neurons so that the value with a larger response becomes relatively larger and suppresses neurons with smaller feedback [225]. The generalization ability of the model is enhanced. Overlapping pooling is used successfully and experiments show that overlapping pooling layers are less prone to overfitting [226]. Data augmentation and dropout are used to suppress overfitting, where data augmentation includes two methods: generating image translations and horizontal reflections and changing the intensity of the RGB channel in the training image [227]. The dropout technique reduces the complex co-adaptations of neurons.

#### 3.2.3. ZF-Net

ZF-Net is a network designed by Zeiler et al. [209] in 2013 with minor improvements based on AlexNet. The functions of the intermediate feature layer and the operation of classification can be understood through novel visualization techniques. Since the size and the stride of the first-layer convolution kernel of AlexNet are too large, the extracted features are mixed with a lot of high-frequency and low-frequency information and lack intermediate-frequency information, so the size of the first-layer filter of ZF-Net is adjusted from 11 × 11 to 7 × 7 and the stride was changed from 4 to 2. This new architecture not only preserves more features in layer 1 and layer 2 but it also improves classification performance.

The deconvnets in ZF-Net are used to map the intermediate layer features to the input pixel space [228], thereby showing the activation patterns contained in the feature map. In ZF-Net, the deconvnets are not used in any learning capability but just as extensions of already trained convolutional networks.

ZF-Net proves that the shallow network learns the image edge, color and texture features, while the deep network learns abstract features of the image and points out the effective reasons and performance improvement methods of the network [229]. At the same time, ZF-Net demonstrates that the deeper network model has better classification robustness under conditions such as image translation and rotation and the higher the level of feature maps, the stronger the feature invariance. Occlusion experiments showed that the ZF-Net model is highly sensitive to the local structure in images rather than just using the broad scene context. During network training, the low-level parameters converge quickly and the higher the level, the longer the training time is required to converge [209].

#### 3.2.4. VGGNet

Simonyan et al. [210] designed the VGGNet and proved that increasing the depth of the network can affect the final performance of the network to a certain extent. By fixing other parameters of the architecture, a network with a very small (3 × 3) convolutional filter at all layers is used for a comprehensive evaluation of the depth-increasing network. Continuously deepening the network structure can improve performance and pushing the depth to 19 weight layers can achieve significant improvements over the existing technical configuration.

The structure of the VGGNet is very consistent and concise [230]. The convolutional kernel size (3 × 3) and max-pooling size (2 × 2) of the same size are used in the whole network. Layers are separated by max-pooling. All the activation units of hidden layers use the ReLU function. An improvement of VGG16 compared to AlexNet is to use several consecutive 3 × 3 convolutional kernels instead of larger convolutional kernels (5 × 5, 7 × 7, 11 × 11) in AlexNet [231]. Specifically, in VGG, a stack of two 3 × 3 convolutional layers replaces one 5 × 5 convolutional layer and a stack of three 3 × 3 convolutional layers replaces one 7 × 7 convolutional layer.

Since VGGNet uses a fixed 3 × 3 convolutional kernel, the increase in the number of layers brings stronger nonlinearity which makes the model’s ability better. Although the number of layers increases, the smaller convolutional kernel reduces the number of parameters in the convolutional layer [232], equivalent to increasing regularization compared with the large convolutional kernel. In this way, under the condition of ensuring the same perceptual field, the depth of the network is improved, and the effect of the neural network is improved to a certain extent.

The VGGNet has three fully connected layers. According to the difference in the total number of convolutional and fully connected layers, six network structures (A, A-LRN, B, C, D, and E) are designed. The famous VGG16 (13 convolutional layers and 3 fully connected layers) and VGG19 (16 convolutional layers and 3 fully connected layers) correspond to D and E, respectively [233]. There is no essential difference between them but the network depth is different. 

In addition to confirming that the classification error decreases as the depth of the ConvNet increases, VGGNet also confirms that deep nets with small filters are better than shallow nets with large filters and that using scale jittering to enhance the training set does help to obtain multi-scale images statistics [210].

#### 3.2.5. GoogLeNet

GoogLeNet is a deep learning structure proposed by Szegedy et al. [211] in 2014. Previous structures such as AlexNet and VGG obtained better training effects by increasing the depth of the network, while GoogLeNet deepened the network (22 layers) and also innovated on the network structure by introducing the Inception structure instead of the traditional operation of simple convolution in combination with the activation function.

Its core idea is to perform multi-scale processing and feature dimensionality reduction; a large number of convolution kernels of different sizes (1 × 1, 3 × 3, and 5 × 5) are used to perform the convolution operation so that features of different sizes are obtained and the feature extraction ability of the single-layer network is enhanced.

GoogLeNet improves the original Inception structure by adding a suitable 1 × 1 convolutional layer before the 3 × 3 and 5 × 5 convolutional layers so that the model parameters can be reduced to a certain extent. The main architecture of GoogLeNet is a 22-layer convolutional neural network stacked with the improved Inception module [234]. At the end of the network, the average pooling was used instead of the fully connected layer [235]. Facts have proved that the top-1 accuracy can be increased by 0.6% [211]. In fact, a fully connected layer was added at the end, mainly for the convenience of fine-tuning in the future. Although the full connections are removed, the network still uses dropout. In order to avoid gradient disappearance, the network additionally adds two auxiliary softmax for forward gradient transmission. During training, their loss is added to the overall loss of the network with a discounted weight. At inference time, these auxiliary networks are removed. All convolutions, including those inside the Inception modules, use rectified linear activations.

#### 3.2.6. ResNet

ResNet was proposed by He et al. [213] in 2015. It is a deep neural network formed by stacking many residual modules. Before ResNet was proposed, all neural networks comprised convolutional and pooling layers. With the superposition of the convolutional layer and the pooling layer, not only does the learning effect become better and better but there will be two problems: one is the vanishing/exploding gradients and the other is the degradation problem [236]. As the number of layers increases, the prediction effect worsens. ResNet solves the problem of vanishing/exploding gradients through data preprocessing and the use of the batch normalization (BN) layer in the network. The residual structure is used to artificially allow some layers of the neural network to skip the connection of neurons in the next layer; the layers are connected to weaken the strong connection between each layer. This approach can solve the degradation problem in deep networks.

The residual structure uses a shortcut connection method, skips one or more layers, and performs simple identity mapping, which neither adds additional parameters nor increases computational complexity, and the entire network can still be trained end-to-end using stochastic gradient descent (SGD) and backpropagation, as shown in Figure 13 [237].

In the residual network structure, assuming that the input is *x*, the learned feature is denoted as *P*(*x*), and the residual that can be learned is denoted as *Z*(*x*); hence, the original learned feature is *P*(*x*) = *Z*(*x*) + *x*. When the residual is *Z*(*x*) = 0, the stacked layer only does the identity mapping at this time and the goal of subsequent learning is to approach the residual to 0 [238].

The biggest difference between an ordinary direct-connected convolutional neural network and ResNet is that ResNet has many bypass branches that connect the input directly to the latter layer so that the latter layer can directly learn the residual [239]. ResNet allows raw input information to be fed directly into the latter layers.

#### 3.2.7. DenseNet

DenseNet was proposed by Huang et al. [217] in 2017, which directly connects all layers (with matching feature-maps sizes) to each other feed-forwardly. For each layer, it uses the feature maps of all previous layers as an input and its own feature maps as an input for all subsequent layers. For a network with *K* layers, the *m*th layer has m inputs, consisting of feature maps from all previous convolutional blocks. Its own feature maps are passed to all *K*−*m* subsequent layers. DenseNet contains a total of $\frac{K\left(K+1\right)}{2}$ connections, which is a dense connection compared to ResNet. Moreover, DenseNet directly connects feature maps from different layers, increasing the input changes of subsequent layers, which can realize feature reuse and improve efficiency.

The network structure of DenseNet is mainly composed of dense blocks and transition layers, as shown in Figure 14. In a dense block, the feature maps of each layer have the same size and can be connected in the channel dimension. Since the input of the latter layer will be very large, the bottleneck layer can be used inside a dense block to reduce the amount of calculation. The transition layer mainly connects two adjacent dense blocks and reduces the size of the feature map.

The DenseNet layers are very narrow; the “collective knowledge” of the network is only added to a small part of feature maps, the rest of the feature maps are kept unchanged, and the final classifier makes a decision according to all the feature maps in the network [217]. The DenseNet architecture clearly distinguishes between information added to the network and information retained. DenseNet requires fewer parameters than traditional convolutional networks because redundant feature maps do not need to be relearned [240].

Each layer of DenseNet can directly access the gradient from the loss function and the original input signal [241], thereby achieving implicit deep supervision and alleviating the phenomenon of gradient disappearance, which helps to train deeper network architectures. Dense connections represent regularization effects, which reduce overfitting on tasks with small training set sizes. Notably, DenseNet can consume a lot of GPU memory if not implemented properly.

#### 3.2.8. MobileNet

MobileNet [214] is an efficient small model based on a streamlined architecture and its core idea is to use depthwise separable convolutions to build lightweight deep neural networks for mobile and embedded vision applications. Two simple global hyperparameters (width and resolution multiplier) are introduced to effectively trade between latency and accuracy. These hyperparameters allow model builders to choose the right size model for their application based on the constraints of the problem.

The basic unit of MobileNet is a depthwise separable convolution which can be decomposed into two parts: a depthwise convolution and pointwise convolution [242], both of which use batchnorm and ReLU nonlinearities. A depthwise convolution is different from a standard convolution. For a standard convolution, its convolution kernel is used on all input channels while a depthwise convolution uses a different convolution kernel for each input channel. In other words, one convolution check corresponds to one input channel [243].

The network is uniformly thinned at each layer by a width multiplier [244]. A resolution multiplier is applied to the input image so that the internal representation of each layer is subsequently reduced by the same multiplier. The width multiplier and resolution multiplier were introduced into MobileNet to reduce the amount of calculation and the number of parameters and to construct a model with a smaller and lower computational cost. The accuracy of MobileNetV1 is slightly lower than that of VGG16 but better than that of GoogLeNet [245]. However, MobileNet has absolute advantages in terms of the calculation and parameter volume.

MobileNetV2 [246] is based on the network architecture of MobileNetV1 and introduces the inverted residual with a linear bottleneck, which significantly improves its accuracy and realizes the excellent effect of multiple image classification and detection tasks for mobile applications. MobileNetV3 [247] combines a hardware network architecture search (NAS) and NetAdapt algorithm to tune to suit mobile phone CPUs and constructs two new MobileNet models: MobileNetV3-Large and MobileNetV3-Small, which are targeted high-resource and low-resource use cases, respectively. Among them, platform-aware NAS searches the global network structure by optimizing each network block and the NetAdapt algorithm is used to search the number of filters in each layer. Looking back at the MobileNet series, the accuracy rate is gradually improving and the delay is also decreasing.

#### 3.2.9. ShuffleNet

Zhang et al. [220] is a neural network structure specially designed for mobile devices with very limited computing power. It mainly uses two technologies: pointwise group convolution and channel shuffle, which greatly reduce the computing overhead while retaining the model accuracy.

In ShuffleNet, a pointwise group convolution and bottleneck structure are used to increase the number of channels without significantly increasing the floating point operations (FLOPs). However, the pointwise group convolution only uses the channel information within the group and does not use the channel correlation information between groups. In order to solve this problem, channel shuffle is introduced to realize the exchange of information between groups to improve accuracy [248].

Given a computational budget, ShuffleNet can use a wider range of feature maps. In ShuffleNet, deep convolution is only performed on bottleneck feature maps to prevent overhead as much as possible. Additionally, ShuffleNet is difficult to implement efficiently on low-power mobile devices.

Since FLOPs cannot accurately estimate the actual runtime, Ma et al. [249] proposed four guidelines for lightweight network design: (i) the same number of input and output channels to minimize memory access costs; (ii) excessive group convolution increase memory access cost; (iii) network fragmentation reduces the degree of parallelism; and (iv) element-wise operations should be reduced. According to the four guidelines, Ma et al. introduced the “channel split” on the basis of ShuffleNet V1 and proposed ShuffleNet V2, which is effective and accurate.

#### 3.2.10. SqueezeNet

SqueezeNet [215] maximizes the computing speed without reducing the accuracy of the model. It can achieve an effect similar to AlexNet on the ImageNet dataset but the number of parameters is 50 times less than that of AlexNet. At the same time, combined with model compression technology, the SqueezeNet model file is about 510 times smaller than AlexNet.

The design of the SqueezeNet architecture mainly adopts three strategies: (i) replace the 3 × 3 filter with a 1 × 1 filter, and the parameters are reduced to 1/9 of the original; (ii) reduce the number of input channels to 3 × 3 filters; and (iii) own sampling is performed at the latter stage of the network so that the convolutional layers have large activation maps and the larger activation maps retain more information, which can provide higher classification accuracy. The first two strategies are about reducing the number of parameters in a CNN while trying to maintain accuracy. The third strategy is to maximize accuracy with a limited parameter budget.

The fire module is the basic building block in SqueezeNet. A fire module consists of two parts: squeeze and expand. Among them, squeeze represents a squeeze layer in the structure of SqueezeNet, which uses a 1 × 1 convolution kernel to convolve the feature maps of the previous layer, and its main purpose is to reduce the dimension of the feature map. The inception structure used by expand consists of a set of continuous 1 × 1 convolutions and a set of continuous 3 × 3 convolutions and then spliced [250].

#### 3.2.11. XceptionNet

The conventional convolution operation maps the cross-channel correlations and spatial correlations of the input feature mapping at the same time. The Inception series structure focuses on decomposing the above process [218] and realizes the decoupling of cross-channel correlation and spatial correlation to a certain extent. The XceptionNet [251] is improved on the basis of InceptionV3, using depthwise separate convolution instead of the traditional Inception block to achieve an entire decoupling of cross-channel correlation and spatial correlation. In addition, the XceptionNet also introduces residual connections.

The XceptionNet architecture consists of 36 convolutional layers structured into 14 modules that constitute the feature extraction library of the network. There are linear residual connections around all modules except the first and last modules. The parameter counts for XceptionNet are similar to those for Inception V3. Compared to Inception V3, XceptionNet has slightly better classification performance on ImageNet datasets and significantly improved on JFT datasets.

#### 3.2.12. U-net

The output of many neural networks is the final classification category label, but on many occasions, such as medical image processing, medical personnel not only want to know the category of the image but also want to know the location distribution of various tissues in the image. U-net can realize the positioning of picture pixels. The network classifies each pixel in the image and the final output is a segmented image according to the category of pixels. U-net was proposed by Ronneberger et al. [212] in 2015. 

It was originally a fully convolutional neural network designed specifically for medical image segmentation. It is now widely used as the main tool for medical image segmentation tasks and has achieved good results. U-net is small in scale compared with many other semantic segmentation networks, so it can also be used for some real-time tasks.

The U-net architecture builds on and expands the network architecture of FCN and its architecture consists of two paths, including a contracting path for capturing context and a symmetric expanding path for precise localization. The contracting path, or the encoder or analysis path, consists of several convolutions, pooling, and down sampling of images, similar to regular convolutional networks and provides classification information. The expanding path, also known as the decoder or synthesis path, combines the high-resolution features from the contraction path with the up sampled output to restore the shape of the original image, giving a per-pixel prediction. The up sampling stage in the U-net architecture adds a lot of feature channels, allowing more original image texture information to be propagated in high-resolution layers. 

U-net has no fully connected layers and is valid for use for convolution throughout. That is, the segmentation map only contains pixels so that the segmentation results can be guaranteed to be based on no missing context features, so the input and output image sizes are not too large same. Using data augmentation with elastic deformation requires only a small number of annotated images.

There have been many variants of U-Net, many of which still continue the core idea, adding new modules or incorporating other design concepts. Zhou et al. [252] summarized the seven improvement mechanisms of U-Net, as shown in Table 11. The characterization and performance of common variation in U-Net are shown in Table 12 and Table 13, respectively.

### 3.3. Advanced Deep Neural Network

#### 3.3.1. Transfer Learning

In medical imaging, datasets are difficult and expensive to build, especially for images related to cancer. Due to factors such as patient privacy and data security, sample data are usually scarce. Medical image annotation is costly and needs to be completed by experienced medical professionals [274]. Transfer techniques have been proven to work well in the field of medical images, for example, using a pre-trained CNN architecture to extract features in an image, feeding them into a fully connected layer, and using average pooling classification to detect and classify breast cancer [275], Zhen et al. [276] verified the feasibility of transfer learning of a neural network-based rectum dose-toxicity prediction model for cervical cancer radiotherapy.

We first need to define two basic concepts to understand the definition of transfer learning. One is “domain”, which is represented by *D*. Domain *D* consists of two parts: feature space *X* and marginal distribution P(X). Therefore, the domain is defined as shown in Equation (8).
(8)D={X, P(X)}

The other is “task”. We denote it by *T*; the task is domain-specific. That is, it makes no sense to talk about a task without the existence of a specific domain. In a specific domain D={X, P(X)}, the task *T* consists of a label space *Y* and a function F(X). F(X) is the prediction function, which can predict the label of the unknown sample *X.* From the perspective of probability, F(X) is approximately equal to P(Y|X), which is the conditional probability distribution of *Y* under the condition of *X*. Thus, we obtain the task definition of the task, as shown in Equation (9).
(9)T={Y, P(Y|X)}

Generally speaking, transfer learning includes multiple source domains to a target domain or a single source domain to a target domain. The most common case is the latter. We strictly define transfer learning, that is, given source domain and its matching task are represented by $D_{S}$ and $T_{S}$ respectively, and target domain and its relevant task are represented by $D_{T}$ and $T_{T}$ respectively, then transfer learning entails acquiring knowledge in $D_{S}$ and $T_{S}$ to help improve the learning of prediction function F(X) in $T_{T}$ [277].

According to the transfer content, transfer learning can be divided into four categories [278]:(i)Instance-based transfer learning entails reusing part of the data in the source domain through the heavy weight method of the target domain learning;(ii)Feature-representation transfer learning aims to learn a good feature representation through the source domain, encode knowledge in the form of features, and transfer it from source domain to the target domain for improving the effect of target domain tasks. Feature-based transfer learning is based on the assumption that the target and source domains share some overlapping common features. In feature-based methods, a feature transformation strategy is usually adopted to transform each original feature into a new feature representation for knowledge transfer [279];(iii)Parameter-transfer learning means that the tasks of the target domain and source domain share the same model parameters or obey the same prior distribution. It is based on the assumption that individual models for related tasks should share a prior distribution of some parameters or hyperparameters. Generally, there are usually two specific ways to achieve this. One is to initialize a new model with the parameters of the source model and then fine-tune it. Secondly, the source model or some layers in the source model are solidified as feature extractors in the new model. Then an output layer is added for the target problem and learning on this basis can effectively utilize previous knowledge and reduce training costs [274];(iv)Relational-knowledge transfer learning involves knowledge transfer between related domains, which needs to assume that the source and target domains are similar and can share some logical relationship, and attempts to transfer the logical relationship among data from the source domain to the target domain.

A brief summary and comparison of the above four types of transfer learning are shown in Table 14.

#### 3.3.2. Ensemble Learning

Ensemble learning (EL) is a class of machine learning framework, not a single machine learning algorithm. It completes learning tasks by building multiple base-learners and combining them with certain strategies [280]. The base learner is usually generated from the training data by a base learning algorithm, which can be a decision tree, a neural network, or another type of machine learning algorithm. Most ensemble methods use a single base learning algorithm to generate a homogeneous base learner. If multiple learning algorithms are used, such as both decision trees and neural networks, a heterogeneous learner is generated [281].

Ensemble learning can alleviate the challenges arising from machine learning, such as class imbalance, concept drift, and the curse of dimensionality. It can be used for classification problem integration, regression problem integration, feature selection integration, outlier detection integration, and many other occasions. The overall prediction performance of the integration is better than that of a single basic learner.

The construction methods of ensemble learning can be divided into two categories: parallel method and serial method [282]. The parallel method refers to the construction of multiple independent learners and taking the average of their predicted results. There is no strong dependence between individual learners and a series of individual learners can be generated in parallel, usually homogeneous weak learners. The representative algorithms are bagging [283] and random forest series algorithms [284]. The serial approach means that multiple learners are built sequentially and there is a strong dependency between individual learners because a series of individual learners need to be generated sequentially, usually heterogeneous learners. Boosting series are representative algorithms, such as AdaBoost [285], gradient lifting [286], etc.

Ensemble learning techniques have been proven to be an effective way to improve the robustness and performance of any medical image classification pipeline [287]. They are also used in cancer diagnosis and prognosis. Brunese et al. [288] proposed a method for identifying the components of an ensemble learner. The ensemble learner was used for the graded detection of brain cancer from magnetic resonance images, with 99% accuracy in detecting benign grade I and malignant grade II, III, and IV brain cancers. Abdar et al. [289] constructed a model called CWV-BANNSVM based on the confidence weighted voting method and the boosting ensemble technique to achieve breast cancer detection with 100% accuracy. Using ensemble learning techniques not only improves the accuracy of cancer detection but also extracts interpretable diagnostic rules. In [290], an ensemble learning model was constructed for the detection of prostate cancer using the stacking technology of random forest classifiers, achieving both accuracy and interpretability.

#### 3.3.3. Graph Neural Network

Graph neural network (GNN) is a framework for directly learning graph-structured data using deep learning [291]. Each node representation is calculated by propagating and aggregating features of adjacent nodes and GNN can capture long-term dependencies between data on graphs [292]. GNN can perform prediction or classification tasks at the level of the entire graph or for each node or edge [293].

GNN can improve the effectiveness of survival prediction for cancer patients. Gao et al. [294] designed a multimodal graph neural network (MGNN) framework to predict cancer survival where multimodal data include clinical data, gene expression, and copy number alteration. MGNN was verified as having strong robustness on four cancer datasets (metastatic colorectal cancer, breast invasive carcinoma, pan-lung cancer, and pediatric acute lymphoid leukemia—Phase II).

In recent years, variants of GNN such as graph convolutional network (GCN), graph attention network (GAT), gated graph neural network (GGNN), graph autoencoder, and graph recurrent network (GRN) have demonstrated excellent performance in many cancer diagnoses.

Interpretability is important for assessing the molecular origin of cancer-related genes, detecting potential artifacts, and increasing trust in modeling methods. GCN can classify unlabeled nodes in the network according to their associated feature vectors and the topology of the network, integrating graph-based data and feature vectors in a natural way. Schulte-Sasse et al. [295] used GCN to develop a machine learning approach known as explainable multi-omics graph integration (EMOGI) to prioritize cancer genes from large datasets. Through their approach, multidimensional multiomics node features, as well as topological features of protein–protein interaction networks, can identify not only highly mutated cancer genes but also genes that contain other types of alterations or interact with other cancer genes. The study fed genomic data from 16 cancer types into the method, identifying 165 new candidate genes that could cause cancer. omicsGAT was introduced for cancer subtype analyses [296]. This approach integrates the graph-based learning and attention mechanism into the mRNA sequencing (RNA-seq) data analysis. The multi-head attention mechanism in omicsGAT can protect the information of specific samples more effectively by assigning different attention coefficients to adjacent samples.

#### 3.3.4. Explainable Deep Neural Network

Explainability refers to “the degree to which an observer can understand the cause of a decision” [297]. Although applying deep learning to cancer detection has achieved remarkable results, neural networks are often regarded as a powerful “black box” with weak explainability. Neural networks are often found to have some unexpected errors in practical use, and explainability may help to discover potential errors and improve the model. In addition, explainability is required based on ethical and legal issues. Especially in the medical field, the explainability of deep learning models is crucial to creating safe and responsible systems suitable for clinical implementation. Explainable artificial intelligence can give an interpretable prediction result based on input data symptoms or medical images to assist doctors in diagnosis [298].

Many interpretability classification methods exist, such as global interpretation vs. local interpretation, Ad-Hoc vs. Post-Hoc, etc. The classifications of interpretability and their characteristics are shown in Table 15 [298].

Explainable deep neural networks (xDNN) [299] constitute a deep learning architecture that synergizes both an inference and reasoning. It is non-iterative and non-parametric which explains its efficiency in terms of time and computational resources. xDNN is based on prototypes, which are actual training data samples (images) with local peaks of the empirical data distribution called typicality as well as of the data density. This generative model is identified in a closed form equivalent to the probability distribution function but derived from the training data fully automatically, without user- or problem-specific thresholds, parameters, or intervention. The numerical Grad-CAM (numGrad-CAM) used in a CNN model can provide an explainability interface for the diagnosis of brain tumors [300]. Windisch et al. [301] trained a neural network to distinguish MRI slices containing vestibular schwannomas, glioblastoma, or no tumor, making the decision of the network easier to interpret, helping to identify potential biases, and selecting appropriate training data.

#### 3.3.5. Vision Transformer

Transformers can develop rapidly in the field of computer vision, mainly due to the following reasons [245,302,303,304]: (i) a strong ability to learn long-distance dependencies. Transformers come with a long dependency feature and use the attention mechanism to capture global context information and extract more powerful features. (ii) Strong multi-modal fusion ability. The input of the transformer does not need to maintain a two-dimensional image. Usually, the initial embedding vector can be obtained by directly operating on the pixels and the information of other modalities can be directly fused at the input end after being converted into a vector. (iii) The model is more explainable. In multi-head attention structure of the transformer, each head applies an independent self-attention mechanism which enables the model to learn relevant information in different representation subspaces for different tasks.

A vision transformer (ViT) migrates the transformer model to computer vision tasks, constituting an image classification method based entirely on the self-attention mechanism [302]. It follows the original design of the transformer as closely as possible. Due to the sequential nature of the video, transformers perform comparably to traditional CNNs and RNNs on video tasks [305].

Since the input of the transformer structure is a two-dimensional matrix, the shape of the matrix can be expressed as (*N*, *D*), where *N* is the length of the sequence and *D* is the dimension of each vector in the sequence. Therefore, in the ViT algorithm, it is first necessary to try to convert the image of *H* × *W* × *C* into the input (*N*, *D*), where (*H*, *W*) is the resolution of the original image and *C* is the number of channels. The specific implementation method is [306,307]:(i)The image of *H* × *W* × *C* is changed into a sequence of *N* × (*P*^2^ × *C*), where P is the size of the image block. This sequence can be viewed as a series of flattened image patches. That is, the image is cut into small patches and then flattened. The sequence contains a total of *N* = *H* × *W*/*P*^2^ image patches and the dimension of each image patch is (*P*^2^ × *C*). After the above transformation, *N* can be regarded as the length of the sequence;(ii)Since the dimension of each image patch is (*P*^2^ × *C*) and the vector dimension we actually need is *D*, we also need to Embed the image patch. That is, each image patch will be linearly transformed and the dimension will be compressed to *D*.

In recent years, the application of a vision transformer in the medical field has attracted the attention of many researchers and it has also been successfully applied to cancer classification based on medical images, including skin cancer [308], prostate cancer [309], colorectal cancer [310], and breast cancer [311], etc.

Based on the traditional vision transformer, a series of ViT improvement methods were proposed for the performance of vision tasks. For example, aiming at the problem that the traditional ViT model does not pay enough attention to the local details of medical images, Zhou et al. [312] designed an adaptive sparse interaction ResNet-ViT dual-branch network (ASI-DBNet) to improve the classification accuracy of brain cancer. The ResNet-ViT parallel structure is designed to simultaneously capture and preserve the local and global information of pathological images and an adaptive sparse interaction block (ASIB) is used to realize the interaction between the ResNet branch and the ViT branch. In [313], a multi-scale visual transformer model for gastric cancer histopathological image detection was constructed which realizes the global automatic detection of gastric cancer images. In this model, the position-encoded transformer model and the convolutional neural network with local convolution are two key modules used to extract global and local information, respectively. In the rapid on-site evaluation (ROSE) technique for the diagnosis of pancreatic cancer, the shuffle instance-based visual transformer (SI-ViT) method [314] was designed to reduce the perturbation between instances and enhance the modeling ability between instances.

### 3.4. Overfitting Prevention Technique

#### 3.4.1. Batch Normalization

In the process of neural network training, because the network parameters are constantly changing according to the gradient descent, the distribution of data in each network layer will change differently. That is, the phenomenon of internal covariate shift appears. Internal covariate shift will cause the input data distribution to shift in a certain direction, resulting in slow network data training. At this time, batch normalization (BN) was introduced to solve the above problems [315]. It is a technique to normalize the activation in the middle layer of the deep neural network, which can greatly improve the speed and accuracy of model training as well as improve network generalization performance [316].

Consider a mini-batch δ of size *k*, $\delta =\left\{x_{1\cdots k}\right\}$. The batch normalization process is shown in Equations (10)–(13).
(10)$$\mu_{\delta }=\frac{1}{k}\sum_{i=1}^{k}x_{i},$$
(11)$$\sigma_{\delta }^{2}=\frac{1}{k}\sum_{i=1}^{k}{\left(x_{i}-\mu_{\delta }\right)}^{2},$$
(12)$$x_{i}^{\Delta }=\frac{x_{i}-\mu_{\delta }}{\sqrt{\sigma_{\delta }^{2}+\tau }},$$
(13)$$BN_{\lambda ,\beta }\left(x_{i}\right)=\lambda x_{i}^{\Delta }+\beta .$$

Here, $\mu_{\delta }$ denotes the mini-batch mean, $\sigma_{\delta }^{2}$ denotes mini-batch variance, *k* denotes the mini-batch size, and $x_{i}^{\Delta }$ denotes the normalization process. τ is a small constant for numerical stability.

In order to maintain the nonlinearity, the parameters λ and β are added into the batch normalization, which is called the SCALE factor and SHIFT factor, respectively. The two parameters are used to scale and shift the input, which is conducive to the coordination of distribution and weight and enhances the expression ability of the network.

The highlight of BN is the normalized activation, which can be applied to any set of activations in the network and integration of this normalization into the network architecture itself. By normalizing activations across the network, it prevents small changes in layer parameters from being amplified as data propagate through the deep network. Normalizing the data can concentrate the data distribution in a range with a large gradient, speed up the convergence of the model, and allow a larger learning rate to be used.

#### 3.4.2. Dropout

Dropout refers to the random dropping of units in a neural network, including both hidden and visible units [162]. The core idea of dropout is randomly dropping units and their connections during deep learning network training which thins the network to prevent complex co-adaptation to the training data, thereby reducing overfitting [317].

Figure 15a is a standard fully connected neural network. Figure 15b results from applying dropout to (a). It randomly drops some neurons at a certain probability.

Notably, deleting a unit only temporarily removes it from the network along with all its incoming and outgoing connections. In general, each unit usually retains a fixed probability, p, independently of others, where p can be chosen using a validation set or simply set to 0.5. For input units, the optimal retention probability is usually closer to 1.

With the need for practical applications, dropout variants have emerged one after another. DropConnect introduced by Wan et al. [318] does not apply dropout directly on neurons but on the weights and biases connecting these neurons. Instead of setting the neuron’s output to 0, the weight or bias of each neuron is set to 0 with a certain probability. Standout, introduced by Ba et al. [319], is a dropout method that tries to improve the standard dropout by adaptively selecting neurons to be dropped according to the value of the weight. Methods to explain dropout from a Bayesian perspective, such as fast dropout [320] and variational dropout [321].

#### 3.4.3. Weight Initialization

The essence of deep learning is to optimize the value of all weights to achieve an optimal solution state. Among them, the layers that need to update the weights include convolutional layers, BN layers, and FC layers. In optimization, weight initialization is important to obtain the optimal solution. 

After weight initialization, the neural network can iteratively update the weight parameter to achieve better performance. In deep learning, the weight initialization method of neural networks has a crucial impact on the convergence speed and performance of the model. If the weight initialization is not appropriate, the model may fall into the local-optimal solution, resulting in the model prediction effect being unideal and even making the loss function oscillate and the model not converge [322]. The most common methods of weight initialization are shown in Table 16.

#### 3.4.4. Data Augmentation

The difficulty in obtaining medical images has always limited it as well as the small amount of data and the high cost of image labeling, which lead to the medical image data being quickly overfitted during the model training process and unable to achieve the best diagnostic results [331,332]. Data augmentation is a technique that expands the training data by using algorithms to augment the size and quality of the training dataset [333].

Data augmentation algorithms in computational vision can be roughly divided into two categories: data augmentation based on basic image processing techniques and data augmentation based on deep learning [334]. Data augmentation methods based on basic image processing techniques mainly include geometric transformations, color space, noise injection, kernel filters, mixing images, random erasing, combination, etc., as shown in Table 17. Data augmentation algorithms based on deep learning mainly include feature space augmentation, adversarial training, GAN-based data augmentation, neural style transfer, meta-learning data augmentations, etc. [333,335,336,337,338], as shown in Table 18. In Table 18, the data augmentation method based on basic image processing technology is denoted as M and the data augmentation algorithm based on deep learning is denoted as D.

## 4. Application of Deep Learning in Cancer Diagnoses

### 4.1. Image Classification

A cancer diagnosis is a classification problem whose nature requires very high classification accuracy. In recent years, deep learning theory has broken through the bottleneck of manual feature selection and has made breakthrough progress in image classification, improving the accuracy of medical image classification and recognition and improving the generalization and promotion capabilities of its applications. CNN is the most successful architecture among deep learning methods. In 1995, Lo et al. [390] applied CNN to medical image analysis.

Fu’adah et al. [391] designed a model that can automatically identify skin cancer and benign tumor lesions using CNN. The model used several optimizers, including RMSprop, Adam, SGD, and Nadam, with a learning rate of 0.001. Among them, the Adam optimizer can identify skin lesions in the ISIC dataset into four categories with an accuracy rate of 99%. DBN was also introduced into the diagnosis of cancer. The detection performance of the DBN-based CAD system for breast cancer is significantly better than that of the traditional CAD system [392]. Anand et al. [393] developed a convolutional extreme learning machine (DC-ELM) algorithm for the assessment of cancer of the knee bone based on histopathological images. The test results showed an accuracy rate of 97.27%. The multi-classifier system based on a deep belief network designed by Beevi et al. [394] can accurately detect mitotic cells, which provides better performance for important diagnosis of breast cancer grading and prognosis.

Shahweli [395] applied an enhancer deep belief network constructed with two restricted Boltzmann machines to identify lung cancer with an accuracy of 96%. For brain tumor detection based on MRI, Kumar et al. [396] used group search-based multi-verse optimization (GS-MVO) to reduce the feature length and handed over the optimally selected features to the DBN to achieve higher classification accuracy. Abdel-Zaher et al. [397] proposed an automatic diagnostic system for detecting breast cancer. The system is based on the DBN unsupervised pre-training stage and then a back-propagation neural network stage with supervision. A pretrained back-propagation neural network with an unsupervised stage DBN achieves higher classification accuracy than a classifier with only one supervised stage. In breast cancer cases, the overall neural network accuracy was increased to 99.68%, the sensitivity was 100%, and the specificity was 99.47%.

Jeyaraj et al. [398] developed an unsupervised generative model based on the DBM for the classification of patterns of regions of interest in complex hyperspectral medical images; the classification accuracy and the success rate are superior to the traditional convolution network. Nawaz et al. [399] designed a multi-class breast cancer classification method based on the CNN model, which can not only classify breast tumors into benign or malignant but can also predict the subclasses of tumors, such as fibroadenoma, lobular carcinoma, etc. Jabeen et al. [400] proposed a breast cancer classification framework from ultrasound images using deep learning and the best-selected features fusion. The experiments were performed on the augmented Breast Ultrasound Image (BUSI) dataset with the best accuracy of 99.1%.

El-Ghany et al. [401] proposed a fine-tuned learning model based on a pretrained ResNet101 network for the diagnosis of multi-types of cancer lesions. A benchmark cancer lesion dataset of over 25,000 histopathology images trained with transfer learning can classify photos of colon and lung cancer into five categories: adenocarcinomas lung, squamous cell carcinomas, benign lung, benign colon, and adenocarcinomas colon. The application results of deep learning in medical image classification are shown in Table 19.

### 4.2. Image Detection

The goal of image detection is to determine where objects are located in a given image (usually with a bounding box) and which class each object belongs to [407]. That is, it includes the two tasks of correct localization and accurate classification. In recent years, object detection algorithms based on deep learning have mainly formed two categories: candidate region-based algorithms and regression-based algorithms. The object detection algorithm based on the candidate region is also called the two-stage method, which divides object detection into two stages: one is to generate a candidate region and the other is to put the candidate region into the classifier for classification and detection.

The object detection algorithm based on candidate regions can obtain richer features and a higher accuracy but the detection speed is relatively slow. Typical algorithms based on candidate regions include R-CNN series, such as R-CNN, Fast R-CNN, Faster R-CNN, Mask R-CNN, etc. In 2014, R-CNN was proposed by Girshick et al. [408]. Although R-CNN has achieved great success in the field of object detection, there are problems, such as the need to train multiple models and a large number of repeated calculations, which leads to the low detection efficiency of R-CNN. In 2015, the team further improved the R-CNN method and proposed the Fast R-CNN model [409]. Fast R-CNN only extracts features from the input image once during the detection process, thereby avoiding the waste of computing resources in feature extraction. However, it still uses selective search to generate proposals, which will also limit the detection speed. In 2015, Ren et al. [410] proposed the Region Proposal Network (RPN) and combined it with Fast R-CNN, which is called the Faster R-CNN network.

In the research on automatic detection and classification of oral cancer, Welikala et al. [411] used ResNet-101 for object classification and Faster R-CNN for object detection. Image classification achieved an F1 score of 87.07% in identifying images containing lesions and 78.30% in identifying images requiring referral. Mask R-CNN [412] is one of the most popular object detection algorithms at present. It abandons the process of pre-determining the region of interest in the traditional two-stage detection algorithm and can directly process the image and then perform network detection on the target area. Therefore, the algorithm can greatly reduce the detection time, and at the same time, the accuracy rate is also greatly improved. 

In [413], Mask R-CNN was used to detect gastric cancer pathological sections, segment cancer fossa, and optimize by adjusting parameters. This method finally made it obtained the test result with an AP value of 61.2 when detecting medical images. In [414], a Mask R-CNN is used to search the entire set of images and detect suspicious lesions, which allows the entire image without prior breast segmentation search and achieves an accuracy of 0.86 on a per-slice basis analysis in the training dataset.

The regression-based object detection algorithm has only one stage and directly regresses the category and coordinates of the target on the input image, and the calculation speed is relatively fast. Typical regression-based algorithms mainly include YOLO series [415], SSD series [416], etc. 

Due to a large number of medical image data modalities and large changes in lesion scales, scholars often improve object detection algorithms in the field of natural images to make them suitable for lesion detection in the field of medical images. Gao et al. [417] used the LSTM model and long-distance LSTM (DLSTM) to collect the time changes of longitudinal data in CT detection of lung cancer. The temporal emphasis model (TEM) in DLSTM supports learning at regular and irregular sampling intervals. Three-dimensional CNN was used to diagnose breast cancer in a weakly supervised manner and locate lesions in dynamic contrast enhancement (DCE) MRI data, showing high accuracy with an overall Dice distance of 0.501 ± 0.274. Studies have shown that the weakly supervised learning method can locate lesions in volumetric radiological images with image-level labels only [418].

Asuntha et al. [419] proposed a CNN method based on the Fuzzy Particle Swarm Optimization (FPSO) algorithm (FPSOCNN) to detect and classify lung cancer. This method greatly reduces the computational complexity of CNN. Shen et al. [420] proposed a globally-aware multiple instance classifier (GMIC) to localize malignant lesions in a weakly supervised manner. The model is capable of processing medical images at the original resolution and is able to generate pixel-level saliency maps, which provide additional interpretability. Applying the model to screening mammography classification and localization requires less memory and is faster to train than ResNet-34 and faster than RCNN.

Ranjbarzadeh et al. [421] proposed C-ConvNet/C-CNN to simultaneously mine local features and global features through two different paths and introduced a new distance attention (DWA) mechanism to consider the influence of tumor center location and brain within the model. In [422], a center-point matching detection network (SCPM-Net) based on 3D sphere representation was proposed, which consisted of two parts: sphere representation and center points matching. The model automatically predicts nodule radius, position, and offsets of nodules without manually designing nodule/anchor parameters. In [423], a two-stage model for breast cancer detection using thermographic images was proposed. In the first stage, VGG16 is used to extract features from the image. In the second stage, the Dragonfly Algorithm (DA) based on the Grunwald-Letnikov (GL) method is used to select the optimal feature subset. The evaluation results show that the model reduces 82% of the features compared with the VGG16 model.

In [424], a dense dual-task network (DDTNet) was used to simultaneously realize the automatic detection and segmentation of tumor-infiltrating lymphocytes (TILs) in histopathological images. A semi-automatic method (TILAnno) was used to generate high-quality boundary annotations for TILs in H- and E-stained histopathology images. Maqsood et al. [425] proposed the transferable texture convolutional neural network (TTCNN), which only includes three layers of convolution and one energy layer. The network integrates the energy layer to extract texture features from the convolutional layer. The convolutional sparse image decomposition method is used to fuse all the extracted feature vectors, and finally, the entropy-controlled firefly method is used to select the best features. The application results of deep learning in medical image detection are shown in Table 20.

### 4.3. Image Segmentation

Image segmentation refers to dividing an entire image into a series of regions [428] which can be considered a problem of pixel-level classification. Accurate medical image segmentation can assist doctors in judging the condition, quantitatively analyzing the lesion area, and providing a reliable basis for correct disease diagnosis. Image segmentation tasks can be divided into two categories: semantic segmentation and instance segmentation [429]. Semantic segmentation assigns a class to each pixel in an image but objects within the same class are not differentiated. Instance segmentation only classifies specific objects. This looks similar to target detection. The difference is that target detection outputs the boundary box and category of the target and instance segmentation outputs the mask and category of the target.

The early traditional medical image segmentation methods mainly include boundary ex-traction, region-based segmentation, threshold-based segmentation, etc. [428]. Among them, NormalizedCut [430], GraphCut [431], and GrabCut [432] are the most commonly used segmentation techniques. With the progress of deep learning networks, a new generation of image segmentation models, such as FCN, U-Net, and their variants, were produced and their segmentation performance was significantly improved.

The image segmentation method based on deep learning establishes the mapping of pixels and the pixels in a certain range to instances or categories through the known sample data. This kind of method uses the powerful nonlinear fitting ability of deep learning and uses a large number of sample data to participate in training. The mapping model established by this method has high accuracy.

FCN is applied to the cancer region segmentation system to fully extract the feature information of different scales in the input image and has a better segmentation effect than the network that does not introduce information of different scales. Recurrent Fully Convolutional Networks (RFCNs) [433] are used to directly solve the segmentation problem in multi-slice MR images.

Wang et al. [434] designed a fully convolutional network which applied multi-parametric magnetic resonance images to detect prostate cancer, including a stage of prostate segmentation and a stage of tumor detection. Dong et al. [435] applied a hybrid fully convolutional neural network (HFCNN) to segment the liver and detect liver metastases. In the system, the loss function is evaluated on the whole image segmentation object. The network processes full images rather than patches, merging different scales by adding links that blend the last detection with lower layers with finer metrics, producing lesion heat maps. Shukla et al. [436] proposed a classification and detection method for liver cancer, which used cascaded fully convolutional neural networks (CFCNs) to train using the input of segmented tumor regions. On 3DIRCAD datasets of different volumes, the total accuracy of the training and testing process is 93.85%, which can minimize the error rate.

The U-Net model [212] is an improvement and extension of FCN. It follows the idea of FCN for image semantic segmentation, that is, using a convolutional layer and pooling layer for feature extraction and then using the deconvolution layer to restore image size. Experiments have proved that the U-Net model can obtain more accurate classification results with fewer training samples. U-Net was developed specifically for medical image data and did not require many annotated images. In addition, due to the existence of high-performance GPU computing, networks with more layers can be trained [437], and the emergence of U-Net in the field of medical image segmentation has been increasing in recent years. 

Ayalew et al. [438] designed a method based on the U-Net architecture to segment the liver and tumors from abdominal CT scan images. The number of filters and network layers of the original U-Net model were modified to reduce the network complexity and improve segmentation performance. Dice scores using this algorithm were 0.96 for liver segmentation, 0.74 for the segmentation of tumors from the liver, and 0.63 for the segmentation of tumors from abdominal CT scan images. However, it still faces great challenges in segmenting small and irregular tumors. The application results of deep learning in medical image segmentation are shown in Table 21.

### 4.4. Image Registration

Medical image registration refers to seeking a kind of (or a series of) spatial transformations for a medical image to make it consistent with the corresponding points on another medical image [460]. This consistency means that the same anatomical point on the human body has the same spatial position in the two matching images. Using the correct image registration method can accurately fuse a variety of information into the same image, making it easier and more accurate for doctors to observe lesions and structures from various angles. At the same time, through the registration of dynamic images collected at different times, the changes in lesions and organs can be quantitatively analyzed, making a medical diagnosis, surgical planning, and radiotherapy planning more accurate and reliable.

To solve the problem of missing data caused by tumor resection, Wodzinski et al. [461] proposed a nonrigid image registration method based on an improved U-Net architecture for breast tumor bed localization. The algorithm works simultaneously at several image resolutions to handle large deformations and a specialized volume penalty is proposed to incorporate medical knowledge of tumor resection into the registration process.

In order to realize the prediction of organ-at-risk (OAR) segmentation on cone-beam CT (CBCT) from the segmentation on planning CT, Han et al. [462] proposed a CT-to-CBCT deformable registration model, which can enable accurate deformation registration CT and CBCT images of pancreatic cancer patients treated with high biological radiation doses. The model uses regularity loss, image similarity loss, and OAR segmentation similarity loss to penalize the mismatch between warped CT segmentation and manually drawn CBCT segmentation. Compared with intensity-based algorithms, this registration model not only improves segmentation accuracy but also reduces the processing time by an order of magnitude. In [463], a novel pipeline was proposed to achieve accurate registration from 2D US to 3D CT/MR. This registration pipeline starts with a classification network for coarse orientation estimation, followed by a segmentation network for predicting ground-truth planes in 3D volumes, enabling fully automated slice-to-volume registration in one shot. In [464], a method for the deformation simulation of inter-fraction in high-dose rate brachytherapy was proposed, which is applied to the deformable image registration (DIR) algorithm based on deep learning, which can directly realize the inter-fraction image alignment of HDR sessions for inter-fraction high dose rate brachytherapy in cervical cancer.

In [465], a CBCT–CBCT deformable image registration was proposed for radiotherapy of abdominal cancer patients. It is based on unsupervised deep learning and its registration workflow includes training and reasoning stages, which share the same feedforward path through a spatial transformation-based network (STN). STNS consists of global generative adversarial networks (GlobalGAN) and local GAN (LocalGAN), which predict coarse and fine-scale motions, respectively. Xie et al. [466] introduced point metric and masking techniques and proposed an improved B-splines-based DIR method to address the large deformation and non-correspondence due to tumor resection and clip insertion to improve registration accuracy. The point metric minimizes the distance between two point sets with known correspondences for regularization of intensity-based B-spline registration. Masking techniques reduce the impact of non-corresponding regions in breast computed tomography (CT) images. This method can be applied to the determination of the target body in the radiotherapy treatment planning after breast cancer surgery. In [467], LungRegNet was proposed for unsupervised deep learning-based deformable image registration (DIR) of 4D-CT lung images. It consists of two subnetworks, CoarseNet and FineNet, which predict lung motion on coarse-scale images and local lung motion on fine-scale images, respectively. The method showed excellent robustness, registration accuracy, and computational speed, with a mean and standard deviation of Target Registration Error (TRE) of 1.00±0.53 mm and 1.59±1.58 mm, respectively. In order to maintain the original topology during deformation to enhance image registration performance, a cycle-consistent deformable image registration called CycleMorph was proposed by [468]. The model can be applied to both 2D and 3D registration and can be easily extended to a multi-scale to solve the memory problem in large-volume registration. The application results of deep learning in medical image registration are shown in Table 22 [469].

### 4.5. Image Reconstruction

Image reconstruction is a method to reconstruct an image based on the data obtained from object detection. Due to the limitations of the physical imaging system, it is difficult for some medical equipment to obtain real medical images and doctors cannot clearly see the specific conditions of lesions in the images, resulting in misdiagnosis, which is not conducive to accurate diagnosis and treatment. Therefore, under the existing hardware conditions, image reconstruction technology can break through the inherent limitations of hardware, improve the quality of medical images, and reduce operating costs but also provide medical personnel with clear images and further improve the accurate diagnosis of diseases. Technology based on deep learning improves the speed, accuracy, and robustness of medical image reconstruction [472,473].

Kim et al. [472] used conventional methods and deep learning-based imaging reconstruction (DLR) with two different noise reduction factors (MRIDLR30 and MRIDLR50) to reconstruct axial T2WI in patients who underwent long-term rectal cancer chemoradiotherapy (CRT) and high-resolution rectal MRI and measured the tumor signal-to-noise ratio (SNR). The results showed that the MR images produced by DLR had a higher resolution and signal-to-noise ratio and that the specificity of identifying pathological complete responses (pCR) was significantly higher than that of conventional MRI. In [474], their results confirmed that the DLIR algorithm for pancreatic protocol dual-energy computed tomography (DECT) significantly improved image quality and reduced the variability of iodine concentration (IC) values compared with hybrid infrared. Kuanar et al. [475] proposed a GAN-based autoencoder network capable of denoising low-dose CT images, which first maps CT images to a low-dimensional manifold and then images are recovered from their corresponding manifold representations.

A major disadvantage of MRI is the long examination time. The deep learning image reconstruction (DLR) method was introduced and achieved good results in scanning acceleration. In the diagnosis of prostate cancer using multiparametric magnetic resonance imaging (mpMRI), Gassenmaier et al. [476] proposed an accelerated deep learning image reconstruction T2-weighted turbo spin echo (TSE) sequence, which reduced the acquisition time by more than 60%. In order to solve the problem that traditional optical image reconstruction methods based on the finite element method (FEM) are time-consuming and cannot fully restore the lesion contrast, Deng et al. [477] proposed FDU-Net, which consists of a fully connected subnetwork, a convolutional encoder–decoder subnetwork, and a U-Net. Among them, the U- Net is used for fast and end-to-end reconstruction of 3D diffuse optical tomography (DOT) images. Training is performed on digital phantoms consisting of randomly located singular spherical inclusions of various sizes and contrasts. The results show that after training, the ability of the FDU-Net to recover the real inclusion contrast and location without using any inclusion information in the reconstruction process is greatly improved and the FDU-Net trained on simulated data can be successfully measured from real patients. Breast tumors were reconstructed from the data.

Feng et al. [478] developed a deep learning-based algorithm (Z-Net) for MRI-guided non-invasive near-infrared spectral tomography (NIRST) reconstruction, which was the first algorithm to use DL for combined multimodality image reconstruction and contributed to better detection of breast cancer. The method avoids MRI image segmentation and light propagation modeling and can simultaneously recover chromophor concentrations of oxy-hemoglobin (HbO), deoxy-hemoglobin (Hb), and water through end-to-end training. Trained neural networks with only simulated datasets can be directly used to distinguish between malignant and benign breast tumors. Wei et al. [479] proposed a real-time 3D MRI reconstruction method from cine-MRI based on unsupervised network for radiation therapy for thoracic and abdominal tumors in MRI-guided radiotherapy for liver cancer. In this method, reference 3D-MRI and cinema-mri are used to generate training data. The application results of deep learning in medical image reconstruction are shown in Table 23.

### 4.6. Image Synthesis

Medical image fusion can be divided into intra-modality and inter-modality image synthesis. Inter inter-modality image synthesis refers to the synthesis of images between two different imaging modalities, such as MR-to-CT, CT-to-MR, PET-to-CT, etc. On the other hand, intra-modality synthesis refers to converting images between two different protocols of the same imaging mode, such as between MRI sequences or restoring images from low-quality protocols to high-quality ones [481]. Deep learning-based methods show higher performance in medical synthetic image accuracy than traditional methods. Deep learning models can more effectively map the correlation between input and output in the design of image transformation and synthesis rules [482] and construct high-level features (such as shapes and objects) by learning low-level features (such as edges and textures) [483] for medical image synthesis with supervised or unsupervised network architectures. Among these architectures, the representative methods include GAN and U-net. 

In [484], a generative model was used to generate new images to reduce dataset imbalances to improve the performance of the automatic gastric cancer detection system. The synthetic network can also generate realistic images even when the lesion image dataset is small. This method allows lesion patches to be attached to various parts of normal images. The experimental results show that the dataset bias is reduced but when the number of synthetic images input to the training dataset is changed, the performance of the model changes. When 20,000 synthetic images were added, the model achieved the highest AP score and when more images were added, the performance of the model decreased. Yu et al. [485] proposed a three-dimensional conditional generation adversarial network (cGAN) and local adaptive synthesis scheme to synthesize fluid-attenuated inversion recovery (FLAIR) images from T1 so as to effectively deal with the single modal brain tumor segmentation based on T1.

Saha et al. [486] proposed TilGAN, an efficient generative adversarial network, to generate high-quality synthetic pathological images of tumor-infiltrating lymphocytes (TILs) and then classify TIL and non-TIL regions. The TilGAN-generated image obtained higher Inception scores and lower initial kernel distances as well as Fréchet Inception distances than the real image. In the classification and diagnosis of skin cancer, Abhishek et al. [487] proposed a conditional GAN-based Mask2Lesion model, which is trained with segmentation masks available in the training dataset and is used to generate new lesion images with any arbitrary mask, augmenting the original training dataset. 

Qin et al. [488] proposed a style-based generation adversarial network (GAN) model to generate skin lesion images in order to solve the problem of a lack of labeled data and an imbalance in dataset categories. This model applies the data augmentation technique based on GANs to the classification of skin lesions and improves the classification performance. Baydoun et al. [489] proposed the sU-cGAN model for MR–CT image translation for cervical cancer diagnosis and treatment. In this model, a shallow U-Net (sU-Net) with an encoder/decoder depth of 2 is used as the generator of a conditional GAN (cGAN) network. The trainable parameters of sU-cGAN are less than those of ordinary cGAN networks.

Zhang et al. [490] used a conditional generative adversarial network to generate synthetic computed tomography images of head and neck cancer patients from CBCT, while maintaining the same cone-beam computed tomography anatomy with accurate computed tomography numbers. The proposed method overcomes the limitations of cone beam computed tomography (CBCT) imaging artifacts and Hounsfield unit imprecision. Sun et al. [491] proposed double U-Net CycleGAN for 3D MR to CT image synthesis. This method solves the problem that GAN and its variants often lead to spatial inconsistencies in contiguous slices when applied to medical image synthesis. Experimental results show that this method can realize the conversion of MRI images to CT images by using unordered paired data and synthesizing better 3D CT images with less computation and memory. Chen et al. [492] used U-net to generate synthetic CT images from conventional MR images in seconds for intensity-modulated radiation therapy treatment planning for the prostate.

Bahrami et al. [493] designed an efficient convolutional neural network (eCNN) for generating accurate synthetic computed tomography (sCT) images from MRI. The network is based on the encoder–decoder network in the U-net model, without softmax layers and the residual network, and the eCNN model shows effective learning ability using only 12 training objects. Synthetic MRI (SyMRI) technology can generate various image contrasts (including fluid-attenuated inversion recovery images, T1WIs, T2-weighted images, and double inversion recovery sequences) and adjust subtle parameter values with information from a single scan to improve the visualization of lesions, which can also quantify T1, T2, and proton density (PD) [494]. In [494], studies have shown that synthetic MRI variables can be used to quantitatively assess bone lesions in the lower trunk of prostate cancer patients and that PD values can be used to determine the viability of prostate cancer bone metastases. Trans-rectal ultrasound (TRUS) images are one of the effective methods to visualize prostate structures. Pang et al. [495] proposed a method to generate TRUS-style images from prostate MR images for prostate cancer brachytherapy. The application results of deep learning in medical image synthesis are shown in Table 24.

## 5. Discussion

Although deep learning has made many breakthroughs in cancer diagnosis based on medical images, there are still some problems or difficulties in applying deep learning in clinical precision diagnosis and personalized treatment. In this section, we will discuss this topic.

### 5.1. Data

#### 5.1.1. Less Training Data

Existing deep learning models have high requirements on the number and quality of images. However, compared with other imaging fields, medical image data are difficult to obtain, especially annotation data, which is reflected in the high cost, cumbersome, and time-consuming annotation [496]; some medical image acquisition methods may even cause physical harm to patients and large amounts of data cannot be naturally obtained [274]. Through literature research, we found that when the deep learning model is applied to the medical image-assisted diagnosis of cancer, the training dataset is relatively small, and the current clinical diagnosis methods entail supervised learning, so insufficient data will inevitably affect the accuracy and stability of diagnosis and prediction. In response to this problem, researchers have designed various solutions, including using data augmentation technology and image fusion to enrich the number of samples, improving the deep learning model, and using the ideas of weak supervision, transfer learning, and multi-task learning to improve the accuracy of classification prediction, and so on, which will be the focus of continuous research.

It is also worth mentioning that GAN-based data augmentation techniques mainly generate medical images from noisy data. Data transformation, oversampling, and adversarial training can be used in combination or multiple data augmentation techniques can be used simultaneously to generate the desired data from the original images. However, overly extensive augmented data can lead to overfitting of the model, potentially worse than before augmentation. In [497], through experimental methods on three different datasets, it was shown that the effectiveness of GANs as a source of medical imaging data is not always reliable, even if the generated images are almost indistinguishable from real data. It is good practice to monitor overfitting during incremental augmentation and define a maximum level of data augmentation based on maximum training accuracy and minimum loss [13].

#### 5.1.2. Class Imbalance

In the clinic, some types of cancer are very common (e.g., adenocarcinoma and ductal carcinoma) while others are less common (e.g., squamous cell carcinoma), which will result in a dataset with an extreme degree of class imbalance [498] and the performance of the classifier is affected. Models trained on such datasets will be biased towards the most prevalent majority class.

Studies have shown that the use of generating new samples [499,500] and class-specialized ensemble technique [498] can balance datasets and improve the classification performance of rare cancer types. In [501], a regularized deep-learning ensemble framework is used to learn from imbalanced multiclass datasets for cancer detection. [502] proposed a deep neural network architecture called VGGIN-Net that includes transfer learning methods, adding batch normalization, flatten, dropout, and dense layers to the network structure, which helps for transferring domain knowledge from a larger ImageNet object dataset to a smaller imbalanced breast cancer dataset. However, the class imbalance in the context of deep learning has not been fully studied and is still a direction for future research.

#### 5.1.3. Image Fusion

Various imaging modes such as CT, PET, MRI, and US have their own advantages and disadvantages. In order to obtain better image effects and bring more valuable diagnostic results to the clinic, fusion imaging technology is used to reorganize and superimpose resources of different imaging modes at the same time, that is, exploit the strengths of all imaging modalities and eliminate or minimize the weaknesses of each mode [503]. A single image generated by fused imaging techniques is more informative and accurate than any single source image and the purpose of image fusion is not only to reduce the amount of data but also to construct images that are more suitable for human and machine perception [504].

Image fusion techniques are mainly classified into six categories: spatial domain fusion, transformation fusion, fuzzy logic, morphological methods, and sparse representation methods [505]. Fusion imaging modes mainly include CT–MRI, MRI–PET, PET–CT, MRI–SPECT, MRI–MRA, Ultrasound–MRI, and X-ray–Ultrasound [506]. Fusion imaging plays an important role in early detection, accurate diagnosis, and scientific treatment of cancer.

Image fusion methods based on deep learning were rapidly developed and widely applied but the existing models lack generalization capabilities and can only effectively fuse specific types of images. Moreover, most current multimodal image fusion on medical images is based on two-modal fusion, which cannot improve the accuracy of tumor contours. The fusion of two or more modalities of medical images is beneficial to the quantification, qualitative, and localization of lesion tissues more clearly.

The latest research [507] introduces a global-local feature extraction strategy and a spatial-frequency fusion strategy to realize three-modal fusion and retain complete texture detail information and global contour information. Meanwhile, a multi-attention mechanism is adopted to extract more effective deep features and more accurate location information. Multi-modal medical image fusion methods need to be paid more attention in the future.

### 5.2. Label

#### 5.2.1. Insufficient Annotation Data

Most deep learning networks, such as CNN, need to label medical images; yet manually labeling medical images is a difficult task. Therefore, minimizing the annotation cost, speeding up the annotation process, and reducing the burden on annotators have become critical challenges in deep learning-based medical image analysis [508]. To address these issues, strategies based on semi-supervised [509,510], self-supervised [511,512], and multi-instance learning [513,514] have attracted extensive attention in recent years. Specific solutions include generating pseudo-labels for unlabeled data, generating more labeled images for model optimization, and learning meaningful patterns from unlabeled image data [515].

Many researchers work on generating pseudo-labels for unlabeled data to enrich training datasets, where the quality of pseudo-labels and the measure of confidence play crucial roles. Self-supervised learning (SSL) is used to extract and learn latent features of large-scale unlabeled datasets without human annotations, which is a promising solution to the challenges posed by the difficulty of large-scale annotation management [516]. In [517], a high-quality classifier from a dataset with only a small fraction of labeled tumor areas is achieved by combining positive and unlabeled learning with domain adaptation techniques.

#### 5.2.2. Noisy Labels

Label noise is unavoidable in many medical image datasets due to a variety of reasons, such as the limited attention or expertise of annotators. Label noise often degrades the performance of deep learning models in medical image-based classification, detection, segmentation, registration, etc. However, few studies directly address the issue of label noise in deep learning for medical image analysis [518]. In the future, learning based on instance-related label noise, multi-label data with label noise, etc., will be an extremely challenging task [519].

#### 5.2.3. Supervised Paradigm

Traditional deep learning-based training models require a large amount of manually labeled data and it is difficult to obtain enough labeled medical images for training, which hinders the further development of traditional supervised learning algorithms. In recent years, researchers have gradually liberated from the traditional paradigm of supervised learning. Paradigms such as weakly supervised learning, semi-supervised learning, and self-supervised learning were concerned and developed [520,521,522,523,524]. In particular, self-supervised learning has achieved great success and is considered a bridge between supervised and unsupervised learning while facilitating the development of weakly supervised and semi-supervised learning [525,526]. However, existing supervised learning strategies for medical images have a lot of room for improvement in terms of performance.

### 5.3. Model

#### 5.3.1. Model Explainability

The explainability of the model is very high in clinical applications. The black box characteristic of the deep learning method itself and the lack of tools to examine the behavior of the black box model are the main reasons hindering its application in the medical field [527]. Images of different cancers vary widely and, while significant research advances have been made in explainable deep-learning cancer detection systems, most of the methods used in the literature are local and post-hoc approaches [528,529,530] and usually explainable models or methods for specific cancer image analysis [298], which are still far from the clinical requirements for the formation of explainable imaging markers. Explainable methods of deep learning models will be a research focus in cancer diagnosis, in addition to the need to include clinicians in the design process of these algorithms and the need to build inherently explainable deep learning models for cancer detection [531].

#### 5.3.2. Model Robustness and Generalization

The goal of deep learning-based cancer diagnosis is to build models that are both robust and generalizable, that is, models that can predict results efficiently and accurately in a variety of situations. Due to the different acquisition, enhancement methods, and modalities of medical images as well as domain shifts on training and test data, there may be many differences in the image performance of the same disease, which significantly affects the performance of clinical cancer diagnosis.

### 5.4. Radiomics

Radiomics is a quantitative medical imaging method and its analysis can be carried out on medical images from different modalities to obtain characteristic data of high throughput of medical images [532,533]. Studies have shown that radiological features are generally not robust to medical image acquisition parameters, such as spatial resolution and image extraction settings [13]. Genomics is a branch of science that studies the structure and function of the entire genome at the molecular level. Studies have shown that molecular/genomic profiling performed after surgery is the most accurate way to assess a patient’s risk and the need for further surgery and adjuvant therapy [98].

By associating genomic data with radiomic features obtained from medical images in multiple dimensions, radiogenomics reveals potential carcinogenic mechanisms, improves cancer diagnosis, non-invasive prediction, and survival prediction at the molecular level [534] and is currently becoming the forefront of precision medicine in oncology. Di Donato et al. [535] showed that preoperative MRI radiomic analysis of patients with endometrial cancer could well predict tumor grade, myometrial invasion, lymphovascular space invasion, and lymph node metastasis. Models based on the radiological characteristics of CT and PET can detect lung nodules, distinguish malignant and benign lesions, and characterize their histology, stage, and genotype [536]. In [537], a time series radiomics predictive model is designed for serial MRI evaluation of standardizing PCa active surveillance. In order to enhance the stability and reproducibility of radiomic features, it is necessary to evaluate and validate feature selection methods and predictive modeling in the future [538].

## 6. Conclusions

This paper reviews and summarizes the research progress of deep learning applied to medical image-based cancer diagnosis. It mainly includes the following aspects: (i) introduces the basic working mechanism of commonly used medical image modalities (CT, MRI, US, X-ray, PET, and histopathology images); (ii) summarizes 9 basic architectures of deep learning and 12 classic Pretrained Models; (iii) introduced five advanced deep neural networks; (iv) sorted out commonly used over-fitting techniques; and (v) 5 applications of deep learning in cancer diagnosis based on medical images are introduced and discussed from three aspects: the data, label, and model.

### 6.1. Limitations and Challenges

Although deep learning technologies and methods have made great progress in cancer diagnosis based on medical images in recent years, research on cancer diagnosis still faces many challenges due to the accuracy and rapidity requirements of cancer diagnosis tasks and the complexity of medical images, which mainly include the following aspects:(i)Datasets problems.

At present, the lack of high-quality datasets in the medical image field has affected the development of deep learning for cancer diagnosis. Due to the lack of data on rare cancers, the current deep-learning research on cancer mainly focuses on common cancers and less attention has been paid to rare cancers.

(ii)The model lacks explainability.

As discussed above, explainability is very important in the medical field. A medical diagnosis system must be transparent, understandable, and explainable to make it easier for clinicians and patients to accept the clinical use of deep learning models [527]. The lack of explainability of the training process and results of the deep learning model seriously hinders its application and promotion as well as hindering the development of intelligent medical diagnosis.

(iii)Poor generalization ability.

Given that the images of different cancers differ greatly, the methods proposed in the current research are mainly limited to the images of different cancers and different imaging types. Therefore, factors such as different equipment, different patient populations, and different imaging parameters cause deep learning to often face a significant performance drop when processing images in new data domains. In order to achieve clinical usability, it is urgent to improve the adaptive and generalization ability of deep learning model.

(iv)Lack of high-performance models for multi-modal images.

Multi-modal data and multi-scale data fusion can improve the accuracy of cancer diagnosis. However, many questions regarding the technical, analytical, and clinical aspects of multimodal fusion remain unresolved. The lack of generalization of the model in cross-center and multi-modal data scenarios also hinders its further development in the medical field.

### 6.2. Future Research Directions

From the challenges discussed above, future research on applying deep learning to a medical image-based cancer diagnosis can start from the following four aspects. (i) The establishment of a growing database of public standards for cancer that is freely available to researchers. (ii) The deep neural network-based pretrained models have the potential to be improved, which can be researched in the aspects of multi-modal data fusion and unsupervised and self-supervised learning. (iii) Vision transformer, integrated learning, etc., are worthy of further study in medical image-based cancer diagnosis. (iv) Few-Shot learning can be further studied in overcoming the small amount of cancer data and the lack of standards.

## Figures and Tables

**Figure 1 cancers-15-03608-f001:**
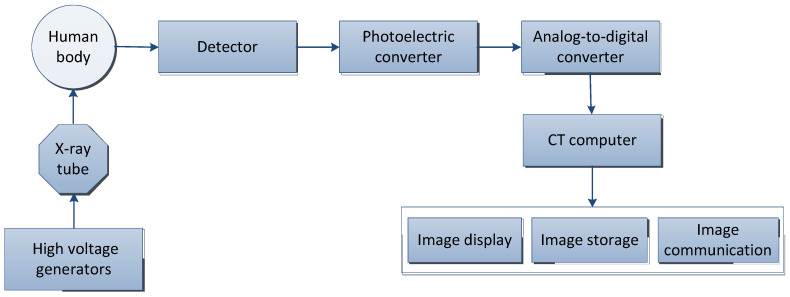
Basic schematic diagram of CT.

**Figure 2 cancers-15-03608-f002:**
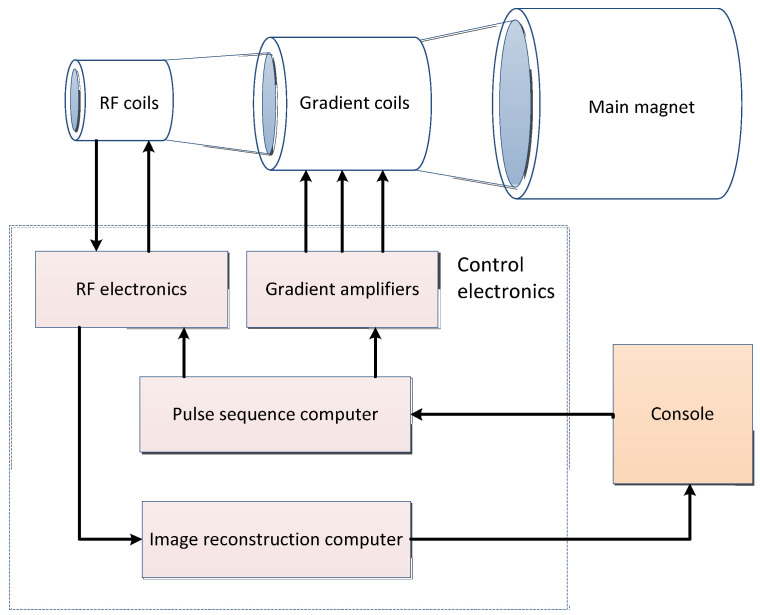
General MRI imaging procedures.

**Figure 3 cancers-15-03608-f003:**
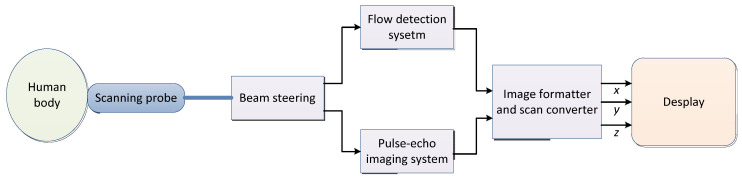
Block diagram of a real-time two-dimensional color flow ultrasonic imaging system.

**Figure 4 cancers-15-03608-f004:**
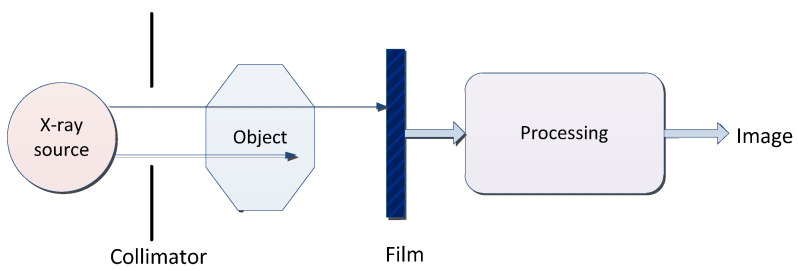
Basic schematic diagram of X-ray.

**Figure 5 cancers-15-03608-f005:**
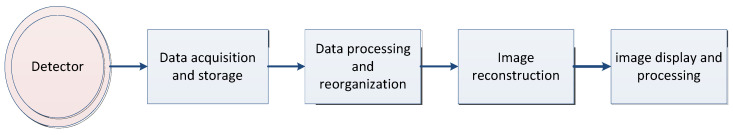
Basic schematic diagram of PET.

**Figure 6 cancers-15-03608-f006:**
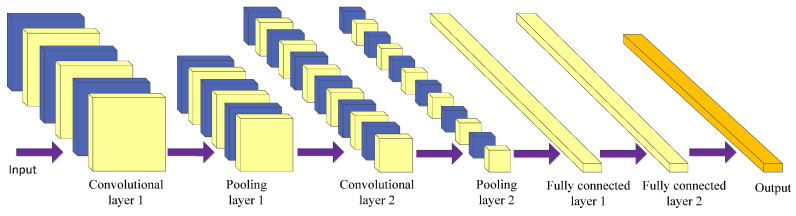
Schematic diagram of convolutional neural network structure.

**Figure 7 cancers-15-03608-f007:**
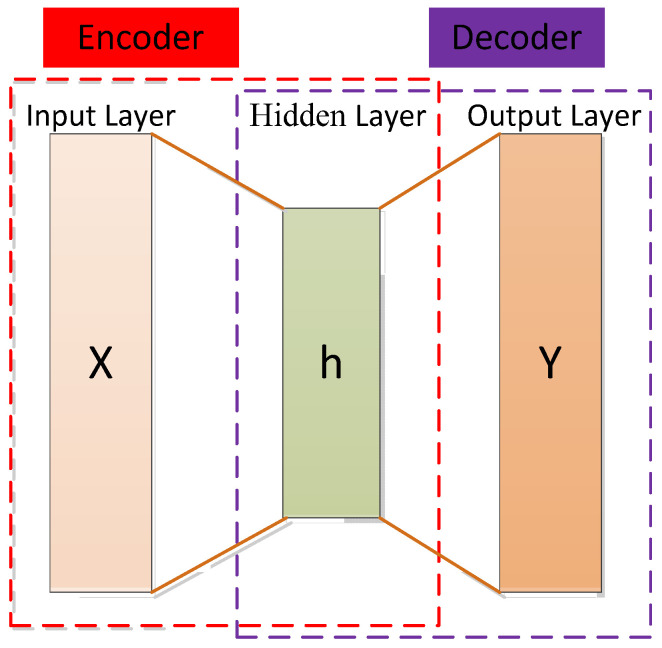
The structure of an autoencoder.

**Figure 8 cancers-15-03608-f008:**
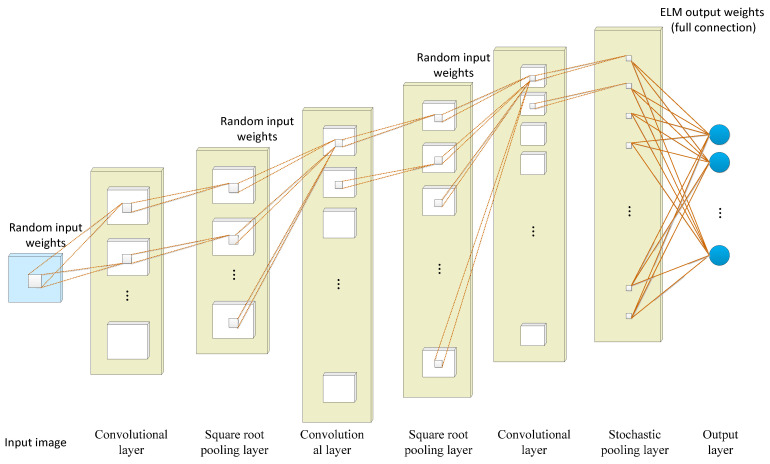
Scheme diagram of the DC-ELM network with three-layer convolution.

**Figure 9 cancers-15-03608-f009:**
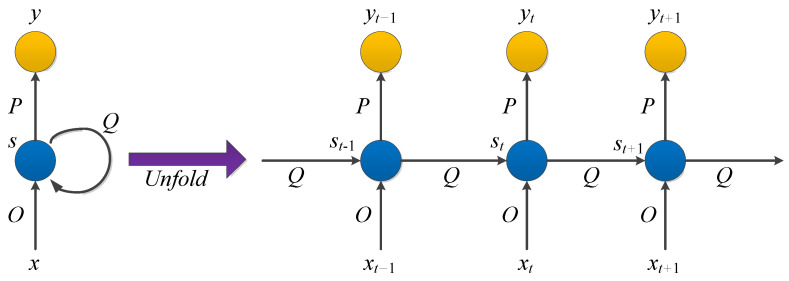
The RNN model structure diagram.

**Figure 10 cancers-15-03608-f010:**
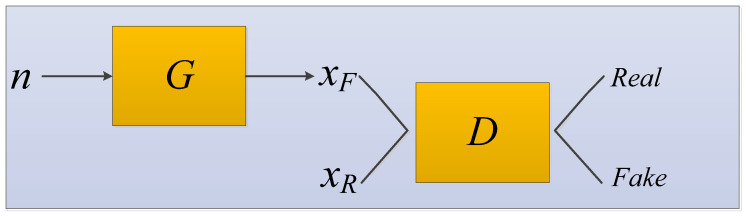
The GAN model structure diagram.

**Figure 11 cancers-15-03608-f011:**
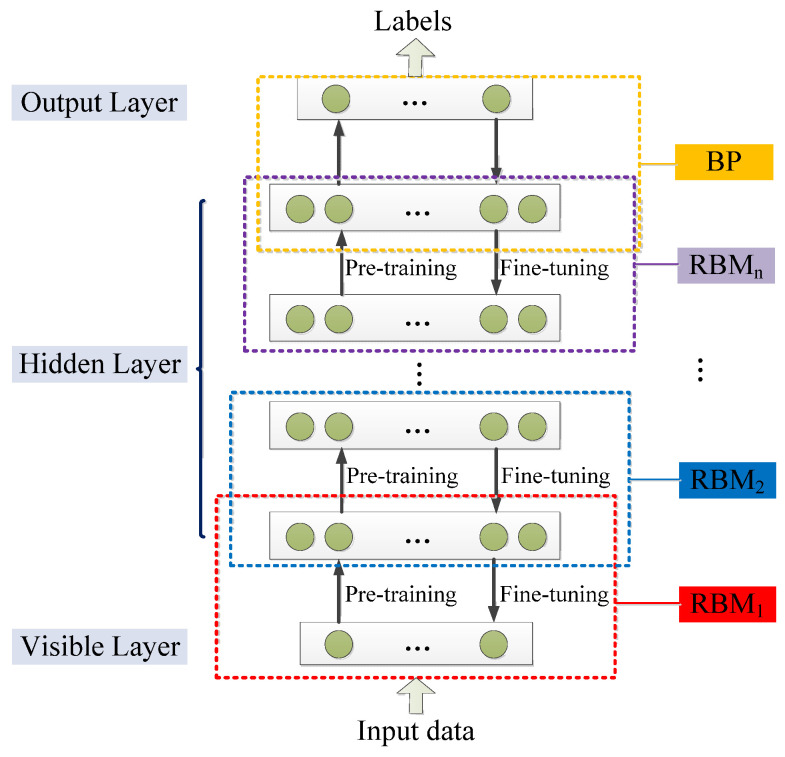
The classic DBN network structure.

**Figure 12 cancers-15-03608-f012:**
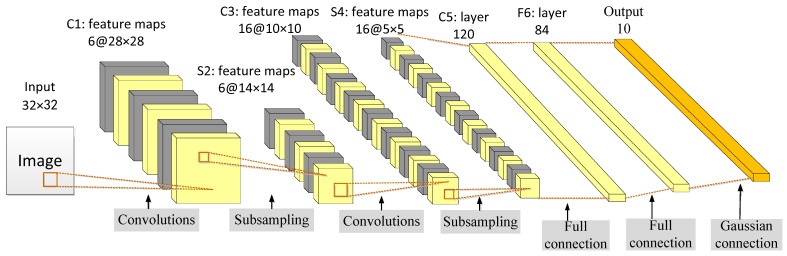
The LeNet-5 architecture.

**Figure 13 cancers-15-03608-f013:**
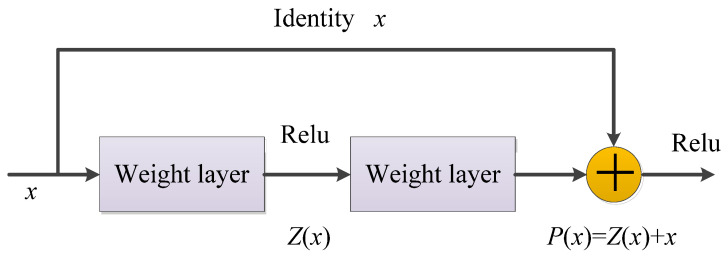
The basic residual block.

**Figure 14 cancers-15-03608-f014:**
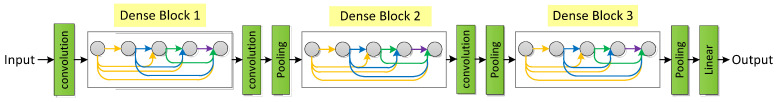
Schematic of a deep DenseNet with three dense blocks.

**Figure 15 cancers-15-03608-f015:**
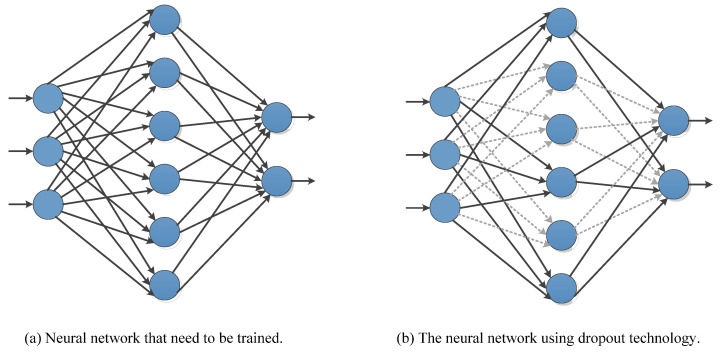
Diagram of applying dropout.

**Table 1 cancers-15-03608-t001:** Comparison of common radiological imaging techniques.

Name	Medium	Imaging Method	Imaging Basis	Features	Advantages	Disadvantages	Radiation	Use Cases
MRI	Magnetic fields and radio waves	Mathematical reconstruction	A variety of parameters	Tomographic images, multi-parameter, multi-sequence imaging, rich grayscale information.	High soft-tissue resolution.	Long examination time, prone to motion artifacts, low spatial resolution, not suitable for patients with metal parts, high price.	No	soft tissue [23], nervous tissue [24], internal organs [25], etc.
X-ray	Ionizing radiation	Transmission projection	Density and thickness	Strong penetrability, wide dynamic range, suitable for image diagnosis with small grayscale differences.	Full picture, real-time, fine image, low cost.	Limited density resolution, overlapping images, poor identification of soft tissues.	Yes	Skeletal system [26], gastrointestinal tract [27], cardiovascular angiography and dynamic observation [28,29], etc.
CT	Ionizing radiation	Mathematical reconstruction	Absorption coefficient	Tomographic image, grayscale image, higher grayscale, can display the tissue density of the human body section.	Fast imaging speed, high-density resolution, no image overlap, further quantitative analysis.	Low spatial resolution, artifacts, and partial volume effects, reflecting only anatomical features.	Yes	Bones [30], lungs, internal organs [31], angiography [32,33], etc.
US	Sound waves	Mathematical reconstruction	Acoustic impedance interface	Suitable for moderate acoustic tissue measurements (soft tissue, muscle, etc.) and imaging of human anatomy and blood flow.	Safe and reliable, no radiation, low cost, can detect very subtle diseased tissue, real-time dynamic imaging.	Poor image contrast, limited field of view, difficult in displaying normal tissue and large lesions.	No	Abdominal organs [34], heart [35,36], ophthalmology [37], obstetrics and gynecology [38], etc.
PET	Radioactive tracer	Mathematical reconstruction	Using positron radionuclide labeling	Concentration image of positrons, showing biological metabolic activity.	High sensitivity, high specificity, whole-body imaging, accurate location can achieve the purpose of early diagnosis.	Low image clarity, poor specificity for inflammation, expensive, the examiner needs to have rich experience.	Yes	Brain blood flow [39], metabolic activity [40], cancer diagnosis [41], etc.

**Table 2 cancers-15-03608-t002:** The basic concepts and features of CT.

CT	Description
Conception	◆ It is one of the most important clinical image inspection tools at present; ◆ It can display invisible lesions to the naked eye and help doctors understand the occurrence and development of diseases, providing great help for clinical diagnosis and treatment; ◆ CT images are layered images; cross-sectional images are commonly used. To visualize an entire organ, multiple sequential slice images are required; ◆ Through the use of image reconstruction programs on CT equipment, coronal and sagittal plane images can also be reconstructed and the relationship between organs and lesions can be viewed from multiple angles.
Feature	◆ CT is superior to traditional X-ray examination because it has high-density resolution and can also achieve axial imaging; ◆ Due to the high-density resolution of CT, soft tissues, bones, and joints can all be seen clearly. CT can better display the organs composed of soft tissues, such as the brain, spinal cord, mediastinum, lung, liver, gallbladder, pancreas, pelvic organs, etc., and display images of lesions on a good anatomical image background.

**Table 7 cancers-15-03608-t007:** The basic concepts and features of the histopathological image.

Histopathological Image	Description
Conception	◆ Histopathology is considered the “gold standard” for cancer diagnosis and clinical decision making; ◆ The histopathological slide preparation process preserves the underlying tissue structure so that some disease features, such as lymphocytic infiltration of cancer, can only be inferred from histopathological images [124]; ◆ During the histopathological evaluation, tissue slides are stained with one or more staining agents to identify the morphological structure of various tissues or abnormal manifestations of material structure [125]; ◆ Hematoxylin–eosin (H and E) staining was the most basic staining method for routine pathological production [126].
Feature	◆ The histopathological images are large in size and high in resolution and the tissue structure in the images is relatively complex; ◆ The feature difference between different categories of images is small and the separability is poor. These characteristics pose challenges for image analysis.

**Table 8 cancers-15-03608-t008:** Comparison of basic architectures of deep learning.

Name	Brief Description	Basic Module
CNN [129]	Feed-forward neural network with convolutional computation and deep structure.	It consists of an input layer, an output layer, and multiple hidden layers. The hidden layers can be divided into convolutional layers, pooling layers, RELU, and fully connected layers.
FCN [130]	Pixel-level classification of images solves the problem of semantic-level image segmentation.	All layers in the model network are convolutional layers.
AE [131]	Efficient feature extraction and feature representation for high-dimensional data using unsupervised learning.	Encoder and decoder.
DC-ELM [132]	Combining the feature abstraction performance of convolutional neural networks and the fast training of extreme learning machines	It consists of an input layer, an output layer, and multiple alternating convolutional and pooling layers.
RNN [133]	A neural network with short-term memory is often used to process time-series data.	It consists of an input layer, a recurrently connected hidden layer, and an output layer.
LTSM [134]	A special kind of RNN capable of learning long dependencies.	A cell, an input gate, an output gate, and a forget gate.
GAN [135]	A deep generative model based on adversarial learning.	Generator and discriminator.
DBN [136]	The number of layers is increased by stacking multiple RBMs to increase expressive power.	Multi-layer RBM.
DBM [137]	A stack of multi-layer RBMs. The middle layer is bidirectionally connected to the adjacent layer.	Boltzmann distribution.

**Table 10 cancers-15-03608-t010:** Classic deep learning architecture.

Name	Year	Brief Description
LeNet-5 [129]	1998	It was designed to solve the problem of handwritten digit recognition and is considered one of the pioneering works of CNN.
AlexNet [208]	2012	The first deep convolutional neural network structure on large-scale image datasets.
ZF-Net [209]	2014	Visualize and understand convolutional networks.
VGGNet [210]	2014	Deeper architecture and simpler form.
GoogLeNet [211]	2014	The introduced Inception combines feature information of different scales to obtain better feature representation.
U-net [212]	2015	Convolutional networks for biomedical image segmentation.
ResNet [213]	2016	The internal residual block uses a skip connection which alleviates the gradient disappearance problem caused by increasing the depth in the deep neural network.
MobileNet [214]	2016	Lightweight CNN for mobile vision problems.
SqueezeNet [215]	2016	Use the fire module to compress parameters.
DarkNet [216]	2016	An open source neural network framework written in C and CUDA.
DenseNet [217]	2017	Re-usage of feature maps.
XceptionNet [218]	2017	Better performed than Inception-v3.
Inception-ResNet [219]	2017	Residual connections were used to increase the training speed of Inception networks.
ShuffleNet [220]	2018	Lightweight CNN using pointwise group convolution and channel shuffle.
NasNet [221]	2018	An architectural building block is searched on a small dataset and then transferred to a larger dataset.
EfficientNet [222]	2019	A compound scaling method with extremely high parametric efficiency and speed.

**Table 11 cancers-15-03608-t011:** Improvement mechanisms of U-Net.

Brief Description	Description	Typical Representative
Dense connection mechanism	There is a direct connection between any two layers.	Dense U-Net [253]; Denseblock U-Net [254]
Residual connection mechanism	The convolution layer of U-Net is replaced with a residual block. The skip connection uses a residual connection path. The encoder and decoder are replaced by a residual network.	Residual U-Net [255]; RRA-UNet [256]
Multi-scale mechanism	Images of multiple scales are input and the results are fused so that the final output combines the features from receptive fields of different sizes.	MU-Net [257]; MDFA-Net [258]
Ensemble mechanism	A group of neural networks processes the same input data in parallel and then combines their outputs to complete the segmentation.	AssemblyNet [259]; DIU-Net [260]
Dilated mechanism	The dilated convolution is used in the encoder and decoder to increase the size of the small convolution kernel while keeping the parameter amount of the convolution unchanged.	DiSegNet [261]; DIN [262]
Attention mechanism	Attention modules are added to the encoder, decoder, and skip connections.	ANU-Net [263]; SA-UNet [264]
Transformer mechanism	The operation of the encoder, decoder, and jump connection is changed to transformer.	TA-Net [265]; FAT-Net [266]

**Table 12 cancers-15-03608-t012:** Characterization of the common variation in U-Net.

Author	Variation	Year	Features
Ronneberger et al. [212]	U-Net	2015	It consists of a contracting path and an expanding path, which are used to obtain contextual information and precise localization, respectively. These two paths are symmetrical to each other.
Çiçek et al. [267]	3D U-Net	2016	The core structure still contains a contracting path and a symmetric expanding path. Overall, 3D volume segmentation is supported and all 2D operations are replaced by corresponding 3D operations, resulting in 3D segmented images that can be segmented with minimal annotation examples.
Oktay et al. [268]	Attention U-Net	2018	It is a hybrid structure that uses attention gates to focus on specific objects of importance and each level in the expansion path has an attention gate through which corresponding features from the contraction path must pass.
Alom et al. [269]	R2U-Net ^1^	2018	It builds on residual and loop techniques, using a simpler concatenation of functions from the encoder to decoder.
Zhou et al. [270]	U-Net++	2018	It is essentially a deeply supervised encoder–decoder network, where the encoder and decoder subnetworks are connected by a series of nested, dense skip pathways, aiming to reduce the semantic gap between feature maps.
Khanna et al. [255]	Residual U-Net	2020	Residuals are integrated into the contraction path of U-Net networks, thus reducing the computational burden and avoiding network degradation problems.
Ibtehaz et al. [271]	MultiResUNet	2020	Res paths are proposed to reconcile two incompatible sets of features. The two feature mappings are more uniform. MultiRes blocks are proposed to augment U-Net’s multiresolution analysis capabilities. It is lightweight and requires less memory.
Li et al. [263]	ANU-Net ^2^	2020	A redesigned dense skip connection and attention mechanism are introduced and a new hybrid loss function is designed.
Zhang et al. [260]	DIU-Net ^3^	2020	The Inception-Res module and the densely connecting convolutional module are integrated into the U-net structure.
Yeung et al. [272]	Focus U-Net	2021	Efficient spatial and channel attention are combined into Focus Gate. Focus Gate uses an adjustable focal parameter to control the degree of background suppression. Short-range skip connections and deep supervision are added. Hybrid focal loss is used to deal with class-imbalanced image segmentation.
Beeche et al. [273]	Super U-Net	2022	In the classical U-Net architecture, a dynamic receiving field module and a fusion up-sampling module are integrated. Image segmentation performance can be improved significantly by dynamic receptive field and fusion up sampling.

^1^ R2U-Net: Recurrent residual CNN-based U-Net; ^2^ ANU-Net: Attention-based nested U-Net; ^3^ DIU-Net: Dense-Inception U-net.

**Table 13 cancers-15-03608-t013:** Performance of common variants of U-Net.

Author	Datasets	IoU (%)	Dice (%)	Precision (%)	Recall (%)	DSC ^1^ (%)	JI ^2^ (%)	Body Part or Organ
Ronneberger et al. [212]	LiTS	89.49	94.45	93.24	95.70	-	-	Liver
	CHAOS	84.46	91.58	89.20	94.08	-	-	Kidney
	CHAOS	76.18	86.48	82.34	91.06	-	-	Spleen
	CHAOS	75.37	85.96	87.31	84.65	-	-	Liver
Çiçek et al. [267]	Private dataset	86.30	-	-	-	-	-	The Xenopus kidney
Oktay et al. [268]	LiTS	93.39	96.58	96.79	96.37	-	-	Liver
	CHAOS	85.77	92.34	90.97	93.76	-	-	Kidney
	CHAOS	84.13	91.38	91.54	91.22	-	-	Spleen
	CHAOS	76.00	86.37	91.11	82.09	-	-	Liver
Alom et al. [269]	LiTS	90.69	95.11	93.80	96.48	-	-	Liver
	CHAOS	85.54	92.21	91.92	92.50	-	-	Kidney
	CHAOS	81.50	89.77	93.60	86.24	-	-	Spleen
	CHAOS	77.80	87.50	92.11	83.39	-	-	Liver
Khanna et al. [255]	LUNA16	-	-	-	98.61 ± 0.14	98.63 ± 0.05	97.32± 0.10	Lung
	VESSEL12	-	-	-	99.61 ± 0.01	99.62 ± 0.003	99.24 ± 0.007	Lung
	HUG-ILD	-	-	-	98.73 ± 0.001	98.68 ± 0.04	97.39 + 0.06	Lung
Zhou et al. [270]	LiTS	94.46	97.15	98.16	96.17	-	-	Liver
	CHAOS	86.58	92.81	90.87	94.82	-	-	Kidney
	CHAOS	81.05	89.53	86.37	92.93	-	-	Spleen
	CHAOS	84.23	91.39	93.06	89.79	-	-	Liver
Yeung et al. [272]	Kvasir-SEG	0.845 (mIoU)	-	91.70	91.60	0.910(mDSC)	-	colorectum
	CVC-ClinicDB	0.893 (mIoU)	-	93.00	95.60	0.941(mDSC)	-	colorectum
Ibtehaz et al. [271]	ISIC-2018	-	-	-	-	-	80.2988 ± 0.3717	Skin lesions
	CVC-ClinicDB	-	-	-	-	-	82.0574 ± 1.5953	Colon
	FMID ^3^	-	-	-	-	-	91.6537 ± 0.9563	U2OS cells; NIH3T3 cells
	BraTS17	-	-	-	-	-	78.1936 ± 0.7868	Glioblastoma; lower grade glioma
Li et al. [263]	LiTS	97.48	98.15	98.15	99.31	-	-	Liver
	CHAOS	90.10	94.79	94.00	95.60	-	-	Kidney
	CHAOS	89.23	94.31	95.19	93.44	-	-	Spleen
	CHAOS	87.89	93.55	94.23	92.88	-	-	Liver
Zhang et al. [260]	KDSB2017 ^4^	-	98.57	-	-	-	-	Lung
	DRIVE + STARE + CHASH_DB1	-	95.82	-	-	-	-	Blood vessel
	MICCAI BraTS 2017	-	98.92	-	-	-	-	Brain tumor
Beeche et al. [273]	DRIVE (n = 40)	-	-	-	-	80.80 ± 0.021	-	Retinal vessels
	Kvasir-SEG (n = 1000)	-	-	-	-	80.40 ± 0.239	-	GI polyps
	CHASE DB1 (n = 28)	-	-	-	-	75.20 ± 0.019	-	Retinal vessels
	ISIC (n = 200)	-	-	-	-	87.70 ± 0.135	-	Skin lesions

^1^ DSC: Dice similarity coefficient; ^2^ JI: Jaccard Indiex; ^3^ FMID: the fluorescence microscopy image dataset developed by Murphy Lab; ^4^ KDSB2017: the Kaggle Data Science Bowl in 2017.

**Table 14 cancers-15-03608-t014:** Comparison of four kinds of transfer learning.

Method	Description	Typical Method	Features
Instance	Samples from different source tasks are collected and reused for learning of the target task.	TrAdaBoost	The method is simple, easy to implement, unstable, and more empirical.
Feature-representation	By introducing the source data features to help complete the learning task of the target data feature domain, the features of the source domain and the target domain are transformed into the same space through feature transformation.	Self-taught learning, multi-task structure learning	Applicable to most methods, the effect is better, it is difficult to solve, and it is prone to overfitting.
Parameter	When some parameters are shared between the source task and the target task, or the prior distribution of model hyperparameters is shared, the model of the source domain is transferred to the target domain.	Learning to learn, Regularized multi-task learning	The similarity between the models can be fully utilized and the model parameters are not easy to converge.
Relational-knowledge	It facilitates learning tasks on target data by mining relational patterns relevant to the target data from the source domain.	Mapping	Compatible for data with dependency and identical distribution

**Table 15 cancers-15-03608-t015:** Comparison of four kinds of xDNN.

Explanation Type	Characteristics
Local	Provide explanations for individual samples.
Global	Provide explanations for a set of samples or the entire model.
Data Modality Specific	Explanation methods that apply only to specific data types.
Data Modality Agnostic	Explanation methods are applicable to any data type.
Ad-Hoc	The model itself is designed to be inherently explainable.
Post-Hoc	Provide explanations after classification is performed.
Model Agnostic	Can explain any model and is not limited to a certain type.
Model Specific	Only available on certain models.
Attribution	Attempts to compute the most important neural network inputs relative to the network result.
Non-Attribution	Develop and validate an explainability method for a given specialized problem.

**Table 16 cancers-15-03608-t016:** The most commonly used weight initialization method.

Method	Year	Goal	Description	Features
Xavier initialization [323]	2010	Solve the problem of gradient disappearance or gradient explosion that may be caused by random initialization.	The range of weight initialization is determined according to the number of input neurons in the previous layer and the number of output neurons in the next layer.	It reduces the probability of gradient vanishing/exploding problems. The influence of the activation function on the output data distribution is not considered and the ReLU activation function does not perform well.
Orthogonal Initialization [324]	2013	Orthogonalize the weight matrix.	It solves the problem of gradient disappearance and gradient explosion under the deep network and is often used in RNN.	It can effectively reduce the redundancy and overfitting in the neural network and improve the generalization ability and performance of the network. Computational complexity is high, so it may not be suitable for large neural networks.
He initialization [325]	2015	The input and output data have the same variance; suitable for neural networks using the ReLU activation function.	In the ReLU network, it is assumed that half of the neurons in each layer are activated and the other half is 0, just divide by 2 on the basis of Xavier to keep the variance constant.	Simple and effective, especially suitable for the case where the activation function is ReLU. Compared with the Xavier initialization, it can effectively improve the training speed and performance of the network. In some cases, it may cause the weight to be too small or too large, thus affecting the network performance.
Data-dependent Initialization [326]	2015	Focus on behavior on smaller training sets, handle structured initialization, and improve pretrained networks.	It relies on the initialization process of the data. All units in the network train at roughly the same rate by setting the network’s weights.	CNN representations for tasks with limited labeled data are significantly improved and representations learned by self-supervised and unsupervised methods are improved. Early training of CNNs on large-scale datasets is greatly accelerated.
LSUV [327]	2015	Produces thin and very deep neural networks.	The weights of each convolutional or inner product layer are pre-initialized with an orthogonal matrix. The variance in the output of each layer is normalized to be equal to 1 from the first layer to the last layer.	It has minimal computation and very low computational overhead. Due to variability in the data, it is often not possible to normalize the variance with the required precision.
Sparse initialization [328]	2017	Achieving sparsity.	The weights are all initialized to 0. Some parameters are chosen randomly with some random values.	The parameters occupy less memory. Redundancy and overfitting in neural networks can be reduced. The generalization ability and performance of the network are improved. Some elements in the weight matrix may be too large or too small, thereby affecting the performance of the network.
Fixup [329]	2019	For networks with residual branches.	Standard initialization is rescaled appropriately.	Deep residual networks can be reliably trained without normalization.
ZerO Initialization [330]	2021	Deterministic weight initialization.	It is based on the identity transformation and the Hadamard transformation and only initializes the weights of the network with 0 and 1 (up to a normalization factor).	Ultra-deep networks without batch normalization are trained. It has obvious characteristics of low-rank learning and solves the problem of training decay. The trained model is more reproducible.

**Table 17 cancers-15-03608-t017:** Common data augmentation methods.

Data Augmentation Category	Advantages	Disadvantages
Geometric transformations	It is simple and easy to implement, which can increase the spatial geometry information of the data set and improve the robustness of the model in different perspectives and positions.	The amount of added information is limited. The data is repeatedly memorized. Inappropriate operations may change the original semantic annotation of the image.
Color space	The method is simple and easy to implement. The color information of the dataset is added to improve the robustness of the model under different lighting conditions.	The amount of added information is limited and repeated memory of the data may change the important color information in the image.
Kernel filter	It can improve the robustness of the model to motion blur and highlight the details of objects.	It is implemented by filtering, which is repeated with the internal mechanism of CNN.
Image Erasing	It can increase the robustness of the model under occlusion conditions, enable the model to learn more descriptive features in the image, and pay attention to the global information of the entire image.	The semantic information of the original image may be tampered with. Images may not be recognized after important partial information is erased.
Mixing images	The pixel value information of multiple images is mixed.	Lack of interpretability
Noise injection	This method enhances the filtering ability of the model to noise interference and redundant information and improves the recognition ability of the model of different quality images.	Unable to add new valid information. The improvement effect on model accuracy is not obvious.
Feature space augmentation	The feature information of multiple images is fused.	Vector data is difficult to interpret.
Adversarial training [339]	It can improve the weak links in the learned decision boundary and improve the robustness of the model.	Extremely slow and inaccurate. More data, deeper and more complex models are required.
GAN-based Data Augmentation	Sampling from the fitted data distribution generates an unlimited number of samples.	A certain number of training samples are required to train the GAN model, which is difficult to train and requires extra model training costs. In most cases, the quality of the generated images is difficult to guarantee and the generated samples cannot be treated as real samples.
Neural style transfer	It can realize mutual conversion between images of the same content and different modalities and can help solve special problems in many fields.	Two datasets from different domains need to be constructed to train the style transfer model, which requires additional training overhead.
Meta learning data augmentations	Neural network is used to replace the definite data augmentation method to train the model to learn better augmentation strategies.	Introducing additional networks requires additional training overhead.
Reinforcement learning data augmentation	Combining existing data augmentation methods to search for the optimal strategy.	The policy search space is large, the training complexity is high, and the calculation overhead is large.

**Table 18 cancers-15-03608-t018:** Typical algorithms of data augmentation based on deep learning.

Data Augmentation Category	Method	Year	Describe	M/D
Geometric transformations	Flipping	-	Usually, the image flip operation is performed about the horizontal or vertical axis.	M
	Rotating	-	Select an angle and rotate the image left or right to change the orientation of the image content.	M
	Zoom	-	The image is enlarged and reduced according to a certain ratio without changing the content in the image.	M
	shearing	-	Move part of the image in one direction and another part in the opposite direction.	M
	translating,	-	Shifting the image left, right, up, or down avoids positional bias in the data.	M
	skew	-	Perspective transforms.	M
	Cropping	-	It is divided into uniform cropping and random cropping. Uniform cropping crops images of different sizes to a set size. Random cropping is similar to translation, the difference is that translation retains the original image size and cropping reduces the size.	M
Color space	Color jittering [213]	2016	Randomly change the brightness, contrast, and saturation of the image.	M
	PCA jittering [340]	2017	Principal component analysis is carried out on the image to obtain the principal component and then the principal component is added to the original image by Gaussian disturbance with mean of 0 and variance of 0.1 to generate a new image.	M
Kernel filter	Gaussian blur filter [341]	1991	The image is blurred using Gaussian blur.	M
	edge filter [342]	1986	Get an image with edge sharpening, highlighting more details of the object.	M
	PatchShuffle [343]	2017	Rich local variations are created by generating images and feature maps with internally disordered patches.	M
Image Erasing	Random erasing [344]	2020	During training, a rectangular region in the image is randomly selected and its pixels are erased with random values.	M
	Cutout [345]	2017	Randomly mask input square regions during training.	M
	HaS [346]	2017	Patches are hidden randomly in training images. When the most discriminative parts are hidden, forcing the network to find other relevant parts.	M
	GridMask [347]	2020	Based on the deletion of regions in the input image, the deleted regions are a set of spatially uniformly distributed squares that can be controlled in terms of density and size.	M
	FenceMask [348]	2020	The “ simulation of object occlusion” strategy is employed to achieve a balance between the information preservation of input data and object occlusion.	M
Mixing images	Mixup [349]	2017	The neural network is trained on convex combinations of pairs of examples and their labels.	M
	SamplePairing [350]	2018	Another image randomly selected from the training data is superimposed on one image to synthesize a new sample, that is, the average value of the two images is taken for each pixel.	M
	Between-Class Learning [351]	2018	Two images belonging to different classes are mixed in a random ratio to generate between-class images.	M
	CutMix [352]	2019	Patches were cut and pasted among the training images, where the ground truth labels was also mixed in proportion to the area of the patches.	M
	AugMix [353]	2019	Using random and diverse augmentation, Jensen–Shannon divergence consistency loss, and mixing multiple augmented images	M
	Manifold Mixup [354]	2019	Using semantic interpolation as an additional training signal, neural networks with smoother decision boundaries at multiple representation levels are obtained.	M
	Fmix [355]	2020	A mixed sample data augmentation using a random binary mask obtained by applying thresholds to low-frequency images sampled from Fourier space.	M
	SmoothMix [356]	2020	Image blending is conducted based on soft edges and training labels are computed accordingly.	M
	Deepmix [357]	2021	Takes embeddings of historical samples as input and generates augmented embeddings online.	M
	SalfMix [358]	2021	A data augmentation method for generating saliency map-based self-mixing images.	M
Noise injection	forward noise adjustment scheme [359]	2018	Insert random values into an image to create a new image.	M
	DisturbLabel [360]	2016	Randomly replace the labels of some samples during training and apply disturbance to the sample labels, which is equivalent to adding noise at the loss level.	M
Feature space augmentation	Dataset Augmentation in Feature Space [361]	2017	Representation is first learned using encoder–decoder algorithm and then different transformations are applied to the representation, such as adding noise, interpolation, or extrapolation.	D
	Feature Space Augmentation for Long-Tailed Data [362]	2020	Use features learned from the classes with sufficient samples to augment underrepresented classes in the feature space.	D
	SMOTE [363]	2002	Interpolate on the feature space to generate new samples.	D
	FeatMatch [364]	2020	A learned feature-based refinement and augmentation method that exploits information from within-class and across-class representations extracted through clustering to produce various complex sets of transformations.	D
Adversarial training [339]	FGSM [365]	2014	This method believes that the attack is to add disturbance to increase the loss of the model and it should be best to generate attack samples along the gradient direction. It is a one-time attack. That is, adding a gradient to a graph only increases the gradient once.	D
	PGD [366]	2017	As the strongest first-order adversary, it is an efficient solution to the internal maximization problem.	D
	FGSM+ random initialization [367]	2020	Eric Wong, Leslie Rice, and J. Zico Kolter. Fast is better than free: Revisiting adversarial training. In ICLR, 2020. Adversarial training using FGSM, combined with random initialization, is as effective as PGD-based training, but at a much lower cost.	D
	GradAlign [368]	2020	Catastrophic overfitting is prevented by explicitly maximizing the gradient alignment inside the perturbation set.	D
	Fast C and W [369]	2021	An accelerated SAR-TR AA algorithm. A network of trained deep encoder replaces the process of iteratively searching for the best perturbation of the input SAR image in the vanilla C and W algorithm.	D
GAN-based Data Augmentation	GAN [194]	2014	sing GAN generative models to generate more data can be used as an oversampling technique to address class imbalance.	D
	CGAN [370]	2014	Add some constraints to GAN to control the generation of images.	
	DCGAN [371]	2015	Combining CNN with GAN, deep convolutional adversarial pair learns representation hierarchies from object parts to scenes in the generator and discriminator.	D
	LapGAN [372]	2015	Images are generated in a coarse-to-fine fashion using a cascade of convolutional networks within the Laplacian Pyramid framework.	D
	InfoGAN [373]	2016	Interpretable representation learning in a completely unsupervised manner via information-maximizing GANs.	D
	EBGAN [374]	2016	An energy-based generative adversarial network model in which the discriminator is treated as a function of energy.	D
	WGAN [375]	2017	The loss function is derived by means of earth mover or Wasserstein distance.	D
	BEGAN [376]	2017	The generator and discriminator are balanced for training an autoencoder-based generative adversarial network.	D
	PGGAN [377]	2017	Generators and discriminators are progressively increasing.	D
	BigGAN [378]	2018	Large-scale GAN training for high-fidelity natural image synthesis based on GAN architecture.	D
	StyleGAN [379]	2019	A style-based GAN generator architecture controls the image synthesis process.	D
	SiftingGAN [380]	2019	The traditional GAN framework is extended to include an online output method for generating samples, a generative model screening method for model sifting, and a labeled sample discrimination method for sample sifting.	D
Neural style transfer	CycleGAN [381]	2017	It only needs to build the respective sample sets of the two image style domains, and unpaired samples can be used for training, which greatly reduces the difficulty of building training sample sets and makes the style transfer between any image domains easier to realize.	D
	Pix2Pix [382]	2017	Conditional adversarial networks as a general solution to image-to-image translation problems.	D
Meta learning data augmentations	Neural augmentation [383]	2017	Before the classification network, an augmented network is introduced to input two randomly selected images of the same category, learn the common content information or style information of the two images through the neural network, and then obtain an “enhanced image”, which is input into the classification network together with the original image for classification model training.	D
	Smart Augmentation [384]	2017	Reduce network losses by creating a network to learn how to generate enhanced data during the training of the target network.	D
Reinforcement learning data augmentation	AutoAugment [385]	2018	The search space is designed and has a strategy consisting of many sub-strategies, the best data augmentation is found by an automatic search strategy.	D
	Fast Autoaugment [386]	2019	The efficient search strategy based on density matching is used to find better data expansion, thus reducing the time of high order training.	D
	Faster AutoAugment [387]	2020	The differentiable policy search pipeline not only estimates the gradient for many conversion operations with discrete parameters but also provides an efficient mechanism for selecting operations.	D
	MARL [388]	2021	An automatic local patch augmentation method based on multi-agent collaboration and the first to use reinforcement learning to find a patch-level data augmentation strategy.	D
	RandAugment [389]	2020	Simplifies the search space, greatly reduces the computational cost of automatic augmentation, and allows the removal of a separate proxy task.	D

**Table 19 cancers-15-03608-t019:** Summary of deep learning for medical image classification.

Reference	Year	Method	Dataset (s)	Imaging Modality	Type of Cancer	Evaluation Metric(s)
Kavitha et al. [402]	2021	BPNN	Mini-MIAS DDSM	Mammography	Breast Cancer	Accuracy: 98.50% Accuracy: 97.55%
Nawaz et al. [399]	2018	DenseNet	BreakHis	histopathological images	Breast Cancer	Accuracy: 95.4%
Spanhol et al. [403]	2016	CNN	BreaKHis	Histopathology	Breast cancer	Accuracy: between 98.87% and 99.34% for the binary classification; between 90.66% and 93.81% for the multi-class classification.
Fu’adah et al. [391]	2020	CNN	ISIC	-	skin cancer	Accuracy: 99%
Anand et al. [393]	2022	DC-ELM	private	histopathological images	Bone Cancer	Accuracy: 97.27% Sensitivity: 98.204% Specificity: 99.568% Precision: 87.832%
Beevi et al. [394]	2017	DBN	MITOS RCC	histopathology images	Breast cancer	F-score: 84.29% F-score: 75%
Shahweli [395]	2020	DBN	IARC TP53	-	lung cancer	Accuracy: 96%
Abdel-Zaher et al. [397]	2016	DBN	WBCD	-	Breast cancer	Accuracy: 99.68% Sensitivity: 100% Specificity: 99.47%
Jabeen et al. [400]	2022	DarkNet-53	BUSI	ultrasound images	Breast cancer	Accuracy: 99.1%
Das et al. [404]	2019	DNN	Private dataset	CT/3D	Liver	Accuracy: 99.38% Jaccard index: 98.18%
Mohsen et al. [405]	2018	DNN	Harvard Medical School website	MRI	Brain	Precision: 97% Rate: 96.9% Recall: 97% F-measure: 97% AUC: 98.4%
El-Ghany et al. [401]	2023	ResNet101	LC25000	histopathological images	Lung and colon cancer	Precision-99.84% Recall: 99.85% F1-score: 99.84% Specificity: 99.96% Accuracy: 99.94%
Attallah [406]	2023	CerCan·Net	SIPaKMeD and Mendeley	-	Cervical cancer	Accuracy: 97.7% (SIPaKMeD) Accuracy: 100% (Mendeley)

**Table 20 cancers-15-03608-t020:** Summary of deep learning for medical image detection.

Reference	Year	Method	Imaging Modality	Type of Cancer	Datasets	Evaluation Metrics
Shen et al. [420]	2021	GMIC	Mammogram	Breast Cancer	NYUBCS + CBIS-DDSM	DSC ^1^ (Malignant): 0.325 ± 0.231 DSC (Benign): 0.240 ± 0.175 PxAP ^2^ (Malignant): 0.396 ± 0.275 PxAP (Benign): 0.283 ± 0.244
Ranjbarzadeh et al. [421]	2021	C-ConvNet/C-CNN	MRI	Brain tumor	BRATS 2018	Dice (mean): 0.9203 (Whole ^3^) 0.9113 (Enh ^4^), 0.8726 (Core ^5^) Sensitivity (mean): 0.9386 (Whole), 0.9217 (Enh), 0.9712 (Core) HAUSDORFF99 (mm): 1.427 (Whole), 1.669 (Enh), 2.408 (Core)
Ari et al. [426]	2018	ELM-LRF	MRI	Brain Cancer	Simulated datasets	Accuracy: 97.18% Sensitivity: 96.80% Specificity: 97.12%
Zhang et al. [414]	2022	Mask R-CNN	MRI	Breast Cancer	DCE-MRI	Accuracy (mean): 0.86 DSC: 0.82
Asuntha et al. [419]	2020	FPSOCNN	CT	Lung cancer	WRT ^6^	Average accuracy: 94.97% Average sensitivity: 96.68% Average specificity: 95.89%
					LIDC	Average accuracy: 95.62% Average sensitivity: 97.93% Average specificity: 6.32%
Zhou et al. [418]	2019	3D CNN	MRI	Breast cancer	Private dataset	Accuracy: 83.7% Sensitivity: 90.8% Specificity: 69.3% Overall dice distance: 0.501 ± 0.274
Welikala et al. [411]	2020	ResNet-101 + Faster R-CNN	MRI	Oral cancer	Private dataset	F1: 87.07% (for identification of images that contained lesions) F1: 78.30% (for the identification of images that required referral)
Zhang et al. [424]	2022	DDTNet	Histopathological image	Breast cancer	BCa-lym	F1: 0.885 Dice: 0.845 PQ ^7^: 0.731 Time: 0.0674
					Post-NAT-BRCA	F1: 0.892 Dice: 0.846 PQ: 0.782 Time: 0.0662
					TCGA-lym	F1: 0.793 Dice: 0.788 PQ: 0.635 Time: 0.0647
Maqsood et al. [425]	2022	TTCNN	Mammogram	Breast cancer	DDSM + INbreast + MIAS	Accuracy: 97.49%
Chattopadhyay et al. [427]	2022	CNN	MRI	Brain cancer	BRATS	Accuracy: 99.74%
Luo et al. [422]	2022	SCPM-Net	3D CT	Lung cancer	LUNA16	Average sensitivity: 89.2%
Cao et al. [413]	2019	Mask R-CNN	Pathological images	Gastric Cancer	Private dataset	AP: 61.2

^1^ DSC: Dice similarity score; ^2^ PxAP: Pixel average precision; ^3^ Whole: Whole tumor; ^4^ Enh: Enhancing tumor; ^5^ Core: Tumor core; ^6^ WRT: Whole real time dataset; ^7^ PQ: Panoptic quality.

**Table 21 cancers-15-03608-t021:** Summary of deep learning for medical image segmentation.

Reference	Year	Method	Datasets	Imaging Modality	Type of Cancer	Evaluation Metrics
Zhu et al. [439]	2018	Adversarial FCN-CRF	Inbreast + DDSM-BCRP	Mammogram	Breast Cancer	Accuracy: 97.0%
Al-Antari et al. [440]	2018	Frcn	Inbreast	Mammogram	Breast Cancer	Dice: 92.69% MCC ^1^: 85.93% Accuracy: 92.97% JSC ^2^: 86.37%
Dong et al. [435]	2020	HFCNN	Private dataset	CT	Liver cancer	Dice: 92%
Shukla et al. [436]	2022	Cfcns	3DIRCAD	CT	Liver cancer	Accuracy: 93.85%
Ayalew et al. [438]	2021	U-Net	3Dircadb01 + LITS	CT	Liver cancer	Dice: 96% (liver segmentation) Dice: 74% (segmentation of tumors from the liver) Dice: 63% (segmentation of tumor from abdominal CT scan images)
Li et al. [441]	2018	CRU-Net	Inbreast DDSM-BCRP	Mammogram	Breast Cancer	Dice Index: 93.66% (Inbreast) Dice Index: 91.43% (DDSM-BCRP)
Shen et al. [442]	2019	Rescu-Net + MS-rescu-Net	Inbreast	Mammogram	Breast Cancer	Dice: 91.78% Jaccard index: 85.12% Accuracy: 94.16%
Li et al. [443]	2019	U-Net + AGS	DDSM	Mammogram	Breast Cancer	Accuracy: 78.38% Sensitivity: 77.89% F-score: 82.24%
Hossain [444]	2019	U-Net	DDSM	Mammogram	Breast Cancer	Dice: 97.80% F-score: 98.50% Jaccard index: 97.4%
Sun et al. [445]	2020	Aunet	INbreast CBIS-DDSM	Mammogram	Breast Cancer	Dice: 79.10% (INbreast) Dice: 81.80% (CBIS-DDSM)
Min et al. [446]	2020	Mask RCNN	Inbreast	Mammogram	Breast Cancer	Dice: 88.00%
Al-Antari et al. [447]	2020	FrCN	Inbreast	Mammogram	Breast Cancer	Dice: 92.69% Accuracy: 92.97% MCC: 85.93% JAC: 86.37%
Abdelhafiz et al. [448]	2020	Vanilla U-Net	Inbreast + DDSM	Mammogram	Breast Cancer	Accuracy (mean): 92.6% IoU ^3^: 90.90%
Rajalakshmi et al. [449]	2020	DS-U-Net	Inbreast CBIS-DDSM	Mammogram	Breast Cancer	Dice: 79% (Inbreast) Sensitivity: 81% (Inbreast) Dice: 82.7% (CBIS-DDSM) Sensitivity: 84.1% (CBIS-DDSM)
Saffari et al. [450]	2020	Cgan	Inbreast	Mammogram	Breast Cancer	Dice: 88.0% Jaccard index: 78.0% Accuracy: 98.0% Precision: 97.85% Sensitivity: 97.85% Specificity: 99.28%
Singh et al. [451]	2020	Cgan	DDSM	Mammogram	Breast Cancer	Dice: 94.0% IoU: 87.0%
Ahmed et al. [452]	2020	Mask RCNN Deep lab v3	MIAS + CBIS-DDSM	Mammogram	Breast Cancer	Average precision: 80.0% (Mask RCNN) Average precision: 75.0% (Deep lab v3)
Bhatti et al. [453]	2020	Mask RCCN-FPN	DDSM	Mammogram	Breast Cancer	Accuracy: 91.0% Precision: 84.0%
Zeiser et al. [454]	2020	U-Net	DDSM	Mammogram	Breast Cancer	Dice Index: 79.39% Accuracy: 85.95% AUC ^4^: 86.40% Sensitivity: 92.32% Specificity: 80.47%
Tsochatzidis et al. [455]	2021	U-Net	DDSM-400 CBIS-DDSM	Mammogram	Breast Cancer	AUC: 89.8% (DDSM-400) AUC: 86.2% (CBIS-DDSM)
Salama et al. [456]	2021	U-Net	MIAS DDSM CBIS-DDSM	Mammogram	Breast Cancer	Dice: 98.87% (DDSM) AUC: 98.88% (DDSM) Sensitivity: 98.98% (DDSM) Precision: 98.79% (DDSM) F1 score: 97.99% (DDSM)
Zhao et al. [457]	2018	Fcnns	BRATS 2013 + BRATS 2015 + BRATS 2016	MRI	Brain	Dice: 86%
He et al. [458]	2022	ANN	LIDC-IDRI	CT	Lung	Accuracy: 94.6% Sensitivity: 95.7% Specificity: 93.5%
Hu et al. [459]	2023	U-Net++	GC-PNI	Pathological images	Gastric cancer	Sensitivity: 97.2% Specificity: 93.3%

^1^ MCC: Matthews correlation coefficient; ^2^ JSC: Jaccard similarity coefficient; ^3^ IoU: Intersection over Union metric; ^4^ AUC: the area under the ROC curve.

**Table 22 cancers-15-03608-t022:** Summary of deep learning for medical image registration.

Reference	Year	Method	Transformation	Supervision	ROI	Modality	Datasets	Evaluation Metrics
Fu et al. [467]	2020	LungRegNet	Deformable	Unsupervised	Lung images	4D-CT	Private dataset + DIRLAB	TRE ^1^ (mean): 1.00 ± 0.53 mm Standard deviation: 1.59 ± 1.58 mm
Kim et al. [468]	2021	CycleMorph	Deformable	Unsupervised	Brain	MRI	OASIS-3	CycleMorph, global: Dice: 0.750 (0.144) Time: 0.01 (GPU) CycleMorph, multi: Dice: 0.756 (0.141) Time: 2.18 (GPU)
					Liver	4D-CT	Private dataset	(1) Arterial → Portal: CycleMorph, global: TRE: 4.722 (3.294) Time: 0.06 (GPU) CycleMorph, multi: TRE: 4.720 (3.275) Time: 0.69 (GPU) (2) Delayed → Portal: CycleMorph, global: TRE: 3.902 (1.694) Time: 0.06 (GPU) CycleMorph, multi: TRE: 3.928 (1.696) Time: 0.69 (GPU)
Xie et al. [466]	2021	IBs DIR ^2^ + PM ^3^ + MT ^4^	Deformable	-	Breast cancer	CT	Private dataset	TRE (mm): 2.27 JD (mean) ^5^: 0.93 ± 0.18
Wodzinski et al. [461]	2021	U-Net	Non-rigid	Semi-Supervised	Breast cancer	CT	Private dataset	TRE (mean): < 6.5 mm RVR ^6^ ≈ 0 Registration time < 1 s
Wei et al. [463]	2021	U-net + ResNet18	Rigid	-	Liver	2D US—3D CT/MR	Private dataset	Distance error: 4.7 ± 1.7 mm Angle errors: 11.6 ± 6.3°
Zhang et al. [470]	2021	GroupRegNet	Deformable	Unsupervised	Lung cancer	4D-CT	DIR-Lab	Average RMSE: 1.48 mm (Landmark300) Average RMSE: 0.85 mm (LandmarkDense)
Salehi et al. [464]	2022	DIRNet	Non-rigid	-	Cervical Cancer	CT	Private dataset	Mean Dice indices: 0.89 ± 0.02 (cervix), 0.96 ± 0.01 (bladder), 0.93 ± 0.02 (rectum) ASSD ^7^: 1.61 ± 0.46 mm (cervix), 1.17 ± 0.15 mm (bladder), 1.06 ± 0.42 mm (rectum) Jaccard coefficient: 0.86 ± 0.04 (cervix), 0.93 ± 0.01 (bladder), 0.88 ± 0.04 (rectum)
Xie et al. [465]	2022	STN	Deformable	Unsupervised	Abdominal cancer	CBCT–CBCT	Private dataset	TRE: 1.91 ± 1.11 mm MAE: 33.42 ± 7.48 HU NCC ^8^: 0.94 ± 0.04 HU
Lei et al. [471]	2022	Dual-feasible framework	Deformable	-	Head-and-neck cancer	pCT-QA CT	Private dataset	MAE ^9^: 40.6 HU PSNR ^10^: 30.8 dB SSIM ^11^: 0.94 TRE: 2.0 mm

^1^ TRE: Target registration error; ^2^ IBs DIR: Intensity-based B-splines DIR; ^3^ PM: Point metric; ^4^ MT: masking technique; ^5^ JD (mean): The mean values of the Jacobian determinant; ^6^ RVR: The relative volume ratio; ^7^ ASSD: Average symmetric surface distance; ^8^ NCC: Normalized cross correlation; ^9^ MAE: The mean absolute error; ^10^ PSNR: Peak-signal-to-noise ratio; ^11^ SSIM: Structural similarity index.

**Table 23 cancers-15-03608-t023:** Summary of deep learning for medical image reconstruction.

Reference	Year	Method	ROI	Modality	Datasets	Evaluation Metrics
Gassenmaier et al. [476]	2021	TSE	Prostate cancer	mpMRI	Private dataset	Acquisition time reduced by more than 60%.
Feng et al. [478]	2022	Z-Net	Breast cancer	MRI + NIRST	Simulation datasets	Number of parameter (M): 3.48 Training time (hours): 3.9 Average MSE ^1^: 0.05±0.04 Average PSNR ^2^: 43.3±3.8 dB Average SSIM ^3^: 0.99
Wei et al. [479]	2022	-	Liver cancer	Cine-MRI	Private dataset	DSC (liver): >96.1% Standard deviation (liver) < 1.3% Localization error (blood vessel): <2.6 mm Standard deviation (blood vessel): <2.0 mm Time: ≈ 100 ms
Koike et al. [480]	2022	U-Net	head and neck cancer	SECT	Private dataset	Lowest MAE: 13.32 ± 2.20 HU Highest PSNR: 47.03 ± 2.33 dB SSIM: 0.9965 ± 0.0009
Deng et al. [477]	2023	FDU-Net	Breast cancer	DOT	Simulated dataset	It has clear advantages over traditional DOT image reconstruction methods, providing over four orders of magnitude speedup in computation time.
Kim et al. [472]	2023	DLR	Rectal cancer	MRI	Private dataset	SNR (MRIDLR30): 9.44 ± 2.31 (*p* < 0.001) SNR (MRIDLR50): 11.83 ± 3.07 (*p* < 0.001) Sensitivity (MRIconv): 48.9% Specificity (MRIconv): 80.8% Sensitivity (MRIDLR30): 48.9% Specificity (MRIDLR30): 88.2%% Sensitivity (MRIDLR50): 38.8% Specificity (MRIDLR50): 86.7%

^1^ MSE: Mean square error; ^2^ PSNR: Peak signal-to-noise ratio; ^3^ SSIM: Structural similarity index measure.

**Table 24 cancers-15-03608-t024:** Summary of deep learning for medical image synthesis.

Reference	Year	Method	ROI	Modality	Datasets	Evaluation Metrics
Chen et al. [492]	2018	U-net	Prostate cancer	MR–CT	Private dataset	Computation time: 3.84–7.65 s MAE ^1^: 29.96 ± 4.87 HU Gamma pass rate (1%/1 mm) > 98.03% Gamma pass rate (2%/2 mm) > 99.36%
Abhishek et al. [487]	2019	Conditional GAN	Skin cancer	Dermoscopic images	ISIC	Achieve an improvement of 5.17% in the mean Dice score as compared to a model trained with only classical data augmentation techniques.
Kanayama et al. [484]	2019	GAN	Gastric cancer	Endoscopic images	Private dataset	The dataset bias was lessened.
Qin et al. [488]	2020	GAN	Skin cancer	Dermatoscopic images	ISIC	Accuracy: 95.2% Sensitivity: 83.2% Specificity: 74.3% Average precision: 96.6% Balanced multiclass accuracy: 83.1%
Bahrami et al. [493]	2020	eCNN	Prostate cancer	MRI-t0-CT	Private dataset	ME ^2^: 2.8 ± 10.3 HU MAE: 30.0 ± 10.4 HU
Saha et al. [486]	2021	TilGAN	TILs ^3^	Pathological images	CGADR ^8^	Accuracy: 97.83%, F1-score: 97.37% Area under the curve: 97%
Baydoun et al. [489]	2021	sU-cGAN	Cervical cancer	MR–CT	Private dataset	Mean MAPE ^4^: 72.83 ± 25.42 Mean RMSE ^5^: 115.74 ± 21.84 PSNR ^6^: 63.41 ± 1.67 SSIM ^7^: 0.839 ± 0.044
Zhang et al. [490]	2022	Conditional GAN	Head and neck cancer	CBCT–sCT	Private dataset	MAE: 36.23 ± 20.24 HU RMSE: 104.60 ± 41.05 HU SSIM: 0.83 ± 0.08 PSNR: 25.34 ± 3.19 dB

^1^ MAE: Mean absolute error; ^2^ ME: Mean error; ^3^ TILs: Tumor-infiltrating lymphocytes; ^4^ MAPE: Mean absolute prediction error; ^5^ RMSE: The root mean square error; ^6^ PSNR: Peak signal-to-noise ratio; ^7^ SSIM: Structural similarity index measure; ^8^ CGADR: The Cancer Genome Atlas data repository.

## Data Availability

The data that support the findings of this study are available from the corresponding author, [Y.Z.], upon reasonable request.

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
