# Peer review of "Deep Learning for Medical Image-Based Cancer Diagnosis"

_cancers, 2023, doi:10.3390/cancers15143608_

Round 1

Reviewer 1 Report (Previous Reviewer 1)

None

Author Response

Answer:

Thank you very much.

Reviewer 2 Report (Previous Reviewer 2)

I read with great interest the Manuscript titled " Deep learning for medical image-based cancer diagnosis”, topic interesting enough to attract readers' attention

Although the manuscript can be considered already of good quality, I would suggest following recommendations:

-       I suggest a round of language revision, in order to correct few typos and improve readability.

-       The authors should expand the “Radiomics” section considering recent evidences about use of radiomics analysis in preoperative management of endometrial cancer. I would be glad if the authors discuss this important point, referring to PMID: 37094971) 

Considered all these points, I think it could be of interest to the readers and, in my opinion, it deserves the priority to be published after minor revisions

I suggest a round of language revision, in order to correct few typos and improve readability.

Author Response

Answer:

Thanks for your suggestion. We revised the language to correct some typos in an effort to improve readability. We expanded the discussion on the "Radiomics" section, and carefully read the article you provided PMID: 37094971, deeply inspired, and cite the article as follows.

‘ 535.      Di Donato, V., E. Kontopantelis, I. Cuccu, L. Sgamba, T.G. D'Augè, et al., Magnetic resonance imaging-radiomics in endometrial cancer: a systematic review and meta-analysis. International Journal of Gynecologic Cancer, 2023: p. ijgc-2023-004313.’

Reviewer 3 Report (Previous Reviewer 3)

The manuscript can be accepted in the current form.

Author Response

Thank you very much.

Reviewer 4 Report (Previous Reviewer 4)

Thanks for addressing all of my comments. I recommend the publication of this review

Author Response

Thank you very much.

This manuscript is a resubmission of an earlier submission. The following is a list of the peer review reports and author responses from that submission.

Round 1

Reviewer 1 Report

The manuscript titled "Deep learning for medical image-based cancer diagnosis" is a comprehensive review and well written. However, there are following concerns

1.        Please cite all facts described in the introduction section. There is only one citation for the text in lines 44-88.

2.       Examinations or techniques? Please use proper words at different places. At some places examination is correct while at other it should be technique.

3.       Please include citations in Table 1 legend.

4.       Common Medical Images? Or imaging techniques.

5.       Section 2 is not relevant. For each technique, describing how it works and what are the usage is of no use. Please describe only relevant text which will be used as terminology in following sections. Table 1 is sufficient for section 2. If the authors want to include more description, please use a table format. But better to remove/cut short the text.

6.       Please cite lines 275-276.

7.       Please cite each one in Table 2.

8.       Please cite text in 282-302, 314-328, 546-554,611-627,638-650, 659-683, 694-605, 719-733, 752-764, 772-786.

9.       Sections 3.1 and 3.2 are written comprehensively, however, most of the techniques have been described using a single citation. Please include more references.

10.   Please cite 1044-1053, 1060-1072, 1155-1164,

The English language is fine and needs minor editing.

Reviewer 2 Report

In my opinion, the analyzed topic is interesting enough to attract the readers’ attention.I think that the abstract of this article is very clear and well structured.

In my opinion, the discussion could be studied in depth and extended. Maybe, it could be useful the evaluation of the radiomics and its new perspectives in diagnosis. In particular, I suggest this article to get deeper in the topic: PMID: 36553988. Because of these reasons, the article should be revised and completed. Considered all these points, I think it could be of interest for the readers and, in my opinion, it deserves the priority to be published after revisions.

 A moderate review should be performed

Reviewer 3 Report

Please consider the following comment:

-The principals of the common methods of medical images should be represented by figures.

Moderate editing of English language required

Reviewer 4 Report

The paper overviews five radiological images, including X-ray, ultra-sound (US), computed tomography (CT), magnetic resonance imaging (MRI), positron emission computed tomography (PET), and histopathological images, are reviewed in this paper. The basic architecture of deep learning and classical pretrained models are comprehensively reviewed. In particular, advanced neural networks emerging in recent years, including transfer learning, ensemble learning (EL), graph neural network, and vision transformer (ViT), are introduced. Five overfitting prevention methods are summarized: batch normalization, dropout, weight initialization, and data augmentation. The application of deep learning technology in medical image-based cancer analysis is sorted out.  The paper is huge (77 pages), It could be split into two or three papers. I suggest focusing on one or two imaging modalities or one or two types of cancer The literature is extensive, however, many important papers are missing. Furthermore, the paper did not mention the criteria for inclusion and exclusion as well as the search strategy to conduct this review article. Also, some comments need to be addressed.

.

Could you please mention the criteria for inclusion and exclusion from the review? Also, how many papers were included and excluded? U searched on google scholar PubMed or what? Your search strategy?

What are the keywords you used in searching strategy?

There should be a section describing the paper's organization by the end of the introduction.

What is the difference between your review and previous reviews regarding the same topic?

Please highlight the novelty and contribution of this review.

I believe that discussing each imaging modality in details does not no adds knowledge, since they are very well techniques. so t recommend dedicating only one section to imaging modalities and briefly discussing each type,

I believe the deep learning section should be split into supervised and unsupervised techniques and discuss each one in a subsection.

EfficientNet, Darknet, NasNet, and InceptionResNet are missing in Table 4

You did not mention any of the deep learning based segmentation methods including  semantic segmentation, Grabcut, U-Net and its extension, and Graphcut segmentation  etc techniques.

The authors did not include several papers for cancer diagnsosis using deep learning . Could you please include the following:

Histo-CADx: duo cascaded fusion stages for breast cancer diagnosis from histopathological images

A Framework for Lung and Colon Cancer Diagnosis via Lightweight Deep Learning Models and Transformation Methods

Ant Colony Optimization-Enabled CNN Deep Learning Technique for Accurate Detection of Cervical Cancer

AI-Based Pipeline for Classifying Pediatric Medulloblastoma Using Histopathological and Textural Images

AUTO-BREAST: A fully automated pipeline for breast cancer diagnosis using AI technology

Hyperparameter optimizer with deep learning-based decision-support systems for histopathological breast cancer diagnosis

A framework for breast cancer classification using multi-DCNNs

CerCan·Net: Cervical cancer classification model via multi-layer feature ensembles of lightweight CNNs and transfer learning

Cervical cancer diagnosis based on multi-domain features using deep learning enhanced by handcrafted descriptors

A multi-task deep learning framework for perineural invasion recognition in gastric cancer whole slide images

Again, the paper is very large in its current state and misleading. I suggest to focus on one or two topics to reduce the paper.